# DistPFN: Test-Time Posterior Adjustment for Tabular Foundation Models under Label Shift

## Abstract

TabPFN has recently gained attention as a foundation model for tabular datasets, achieving strong performance by leveraging in-context learning on synthetic data. However, we find that TabPFN is vulnerable to *label shift*, often overfitting to the majority class in the training distribution. To address this limitation, we propose *DistPFN*, the first test-time posterior adjustment method designed for in-context tabular foundation models. DistPFN rescales predicted class probabilities by downweighting the influence of the training prior (i.e., the class distribution of the context) and emphasizing the contribution of the model's predicted posterior, without modifying the architecture or requiring additional training. We further introduce *DistPFN-T*, which incorporates temperature scaling to adaptively control the adjustment strength based on the discrepancy between prior and posterior. We evaluate our methods on over 250 OpenML datasets, demonstrating substantial improvements for various TabPFN-based models in classification tasks under label shift, while maintaining strong performance in standard settings without label shift.

## 1 Introduction

Tabular data is among the most prevalent data formats across various domains, such as healthcare (Johnson et al., 2016) and finance (Arun et al., 2016). Tree-based models (Chen & Guestrin, 2016; Ke et al., 2017) have consistently demonstrated strong performance on tabular tasks, owing to their ability to handle heterogeneous feature types with minimal hyperparameter tuning. Recently, deep learning (DL) methods, especially transformer-based models (Huang et al., 2020; Gorishniy et al., 2021), have emerged as strong alternatives by capturing complex feature interactions.

Among these methods, TabPFN (Hollmann et al., 2023) introduces in-context learning (ICL) to tabular classification by pretraining on synthetic datasets and producing predictions for test samples in a single forward pass. While TabPFN achieves strong performance on small-scale datasets, it suffers from scalability issues due to the quadratic complexity of self-attention (Vaswani et al., 2017). To address this limitation, several extensions have been proposed to improve inference efficiency on larger datasets (Thomas et al., 2024; Xu et al., 2025; Zeng et al., 2025).

In this paper, we highlight an overlooked limitation of TabPFN, namely its vulnerability to *label shift*, which is a critical scenario in tabular learning and frequently arises in real-world tasks (Kim et al., 2024). We observe that TabPFN tends to overfit to the majority class in the training dataset (i.e., *majority-class bias*), resulting in poor performance when the class distribution in the test dataset differs. As shown in Figure 1 and Table 1, TabPFN-v2 (Hollmann et al., 2025) exhibits a strong majority-class bias, making incorrect predictions even when trained and tested on the *same* dataset.

To this end, we propose *DistPFN*, a simple yet effective test-time adaptation method that improves the robustness of TabPFN-based models to label shift, without modifying the architecture or updating any parameters. Specifically, it adjusts the model's output distribution (i.e., **posterior**) by reweighting class probabilities based on the ratio between the posterior and the class distribution of the training dataset (i.e., **prior**). Intuitively, this adjustment downweights the influence of the training distribution and amplifies the impact of the observed test samples. Furthermore, to make the adjustment more adaptive to distributional mismatch between the labels of the training and test datasets, we propose *DistPFN-T*, which adjusts the reweighting intensity via temperature scaling, where the temperature is determined based on the discrepancy between the posterior and the prior. Unlike classical correction

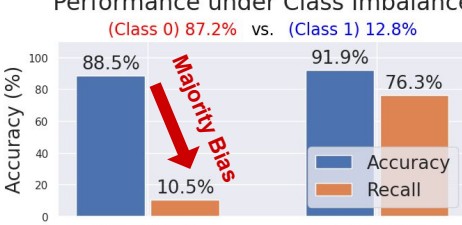

Figure 1: **Majority-class bias of TabPFN.** TabPFN suffers from majority-class bias, resulting in poor recall for the minority class.

| TabPFN-v2 (Nature 2025) | | | + DistPFN (Ours) | | |
|---|---|---|---|---|---|
| (%) | $\hat{y} = 0$ | $\hat{y} = 1$ | (%) | $\hat{y} = 0$ | $\hat{y} = 1$ |
| $y = 0$ | **87.2** | 0.0 | $y = 0$ | 81.1 | 6.1 |
| $y = 1$ | **11.1** | 1.7 | $y = 1$ | 3.0 | 9.8 |
| Total | **98.3** | 1.7 | Total | 84.1 | 15.9 |

Table 1: **Confusion matrices.** TabPFN exhibits severe *majority-class bias*, predicting **98.3%** of samples as the **majority class**, whereas DistPFN alleviates this issue through a simple *test-time adjustment*.

methods that require explicit estimation of test priors, our approach is novel in that it leverages the in-context inference structure to enable an adjustment without additional training or prior estimation.

We conduct extensive experiments on over 250 classification datasets to evaluate the effectiveness of DistPFN in both standard and label-shifted classification scenarios. As shown in Figure 2, DistPFN significantly improves accuracy of TabPFN-v2 (Hollmann et al., 2025) under label shift, with the x-axis indicating the degree of shift (see Sec. 4.4) and the y-axis showing average accuracy over 253 datasets. Our main contributions are summarized as follows:

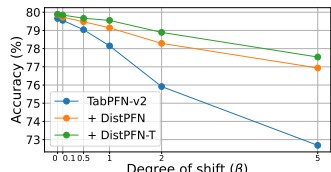

Figure 2: Robustness to shift.

- We identify an overlooked limitation of TabPFN—its vulnerability under *label shift*, as it tends to overfit to the majority class in training dataset. The performance of various TabPFN-based models degrades drastically as the degree of shift increases.
- We propose **DistPFN**, a novel and simple test-time adaptation method that adjusts TabPFN's output distribution based on the ratio between the predicted posterior and the prior representing the class distribution of the training dataset. We further introduce **DistPFN-T**, which extends this approach by applying temperature scaling to adaptively control the strength of adjustment according to the degree of distributional mismatch between the prior and the posterior.
- We present extensive evaluations on over 250 classification datasets, demonstrating that our methods significantly improve the performance of various TabPFN-based models under label shift, surpassing baseline methods and achieving state-of-the-art (SoTA) performance.
- We provide a theoretical interpretation of our method, showing that it can be viewed from both 1) classical label shift correction and 2) Bayesian inference, with details provided in Appendix D.

## 2 RELATED WORKS

**Gradient Boosting Decision Trees (GBDTs).** GBDTs (Chen & Guestrin, 2016; Ke et al., 2017; Prokhorenkova et al., 2018) are widely used for tabular data due to their strong inductive biases, minimal preprocessing, and robust performance across diverse datasets (Grinsztajn et al., 2022; McElfresh et al., 2023). Despite advances in deep learning, GBDTs remain dominant in tabular tasks, as gradient-based models struggle to encode suitable inductive biases and often underperform in several benchmarks (Grinsztajn et al., 2022; Shwartz-Ziv & Armon, 2022; McElfresh et al., 2023).

**Tabular Deep Learning (DL).** Transformer-based models have been proposed to capture complex feature interactions in tabular data, where TabTransformer (Huang et al., 2020) applies contextual embeddings to categorical features using self-attention. Several other architectures have also been introduced (Arik & Pfister, 2021; Somepalli et al., 2021), but these methods typically require extensive hyperparameter tuning and often fail to generalize across datasets (Kadra et al., 2021; Grinsztajn et al., 2022). Recently, TabM (Gorishniy et al., 2024a) proposes an MLP-based ensemble model that generates multiple predictions per instance with shared parameters, and ModernNCA (Ye et al., 2024) introduces a differentiable $k$NN approach based on neighborhood components analysis. RealMLP (Holzmüller et al., 2024) simplifies MLPs with improved design and meta-tuned default parameters, and TabR (Gorishniy et al., 2024b) augments inputs by retrieving similar training examples.

**Tabular Foundation Models.** TabPFN (Hollmann et al., 2023) is a foundation model for tabular classification that uses in-context learning (ICL) via large-scale synthetic pretraining. It predicts test instances by conditioning on training inputs and labels, without gradient updates. TabPFN-v2 (Hollmann et al., 2025) enhances scalability and generalization through dual-axis attention over samples and features, and structurally diverse synthetic pretraining. Limited by its computational

Figure 3: **Overall framework of DistPFN.** (a) **TabPFN** exhibits a *majority-class bias* under label shift, predicting test instances toward the dominant class in the training dataset. (b) **DistPFN** mitigates this bias via a simple *test-time adaptation* method that rescales the predicted class probabilities for each test instance. This scaling factor is computed from the 1) the training class distribution (i.e., prior, $p_{\text{train}}(y)$) and 2) the predicted class distribution of the test instances (i.e., posterior, $p_{\text{test}}(y)$).

complexity, TabPFN has led to several extensions, where LoCalPFN (Thomas et al., 2024) improves TabPFN by retrieving neighbors of test samples and fine-tuning on this local context, and MixturePFN (Xu et al., 2025) scales TabPFN to larger datasets by combining nearest-neighbor sampling with bootstrapped fine-tuning at inference time. TuneTable (Feuer et al., 2024) scales TabPFN to larger datasets by learning dataset-specific contexts through fine-tuning, and TabFlex (Zeng et al., 2025) replaces softmax attention of TabPFN with linear attention to improve efficiency. TabICL (Qu et al., 2025) uses a two-stage architecture with column-then-row attention to embed rows.

**Label shift.** Several studies have addressed label shift by rescaling classifier outputs, typically requiring estimation of the test distribution (Elkan, 2001; Lipton et al., 2018; Azizzadenesheli et al., 2019). In contrast, our method avoids test prior estimation and instead leverages only the training prior (e.g., the in-context dataset) and the predicted distribution, yielding a simple and efficient plug-in adjustment that requires no architectural modifications. Moreover, unlike Drift-Resilient TabPFN (Helli et al., 2024), which addresses temporal shift through causal pretraining, our method specifically addresses label shift and enables test-time adaptation of pretrained models without retraining.

## 3 PRELIMINARIES

**Tabular classification.** A tabular dataset consists of instances $x_i \in \mathbb{R}^d$, where each $x_i$ is a $d$-dimensional feature vector composed of numerical, ordinal, or (one-hot encoded) categorical attributes. Each instance is associated with a label $y_i \in \{1, \ldots, C\}$ indicating one of $C$ predefined classes. The training dataset is denoted as $\mathcal{D}_{\text{train}} = \{(x_i, y_i)\}_{i=1}^{N_{\text{train}}}$, and the test dataset as $\mathcal{D}_{\text{test}} = \{x_j\}_{j=1}^{N_{\text{test}}}$. In a tabular classification task, a model $f$ predicts the class labels $y_j$ given the $x_j$ from the test set.

**Tabular ICL.** TabPFN-based methods (Hollmann et al., 2023; 2025) predict test labels by conditioning on the labeled $\mathcal{D}_{\text{train}} = \{(x_i, y_i)\}_{i=1}^{N_{\text{train}}}$ and the unlabeled $\mathcal{D}_{\text{test}} = \{x_j\}_{j=1}^{N_{\text{test}}}$. Note that these models infer the corresponding test labels $y_j$ in a single forward pass without any gradient updates.

**Label shift.** In this paper, we aim to address label shift, which is a distribution shift scenario where the marginal label distribution differs between training and test datasets, i.e., $p_{\text{train}}(y) \neq p_{\text{test}}(y)$. Specifically, for each class $c_k \in \{1, \ldots, C\}$, the label distributions are estimated as

$$p_{\text{train}}(y = c_k) = \frac{|\{i \in [N_{\text{train}}] : y_i = c_k\}|}{N_{\text{train}}}, \quad p_{\text{test}}(y = c_k) = \frac{|\{j \in [N_{\text{test}}] : y_j = c_k\}|}{N_{\text{test}}},$$

where $p_{\text{test}}(y = c_k)$ cannot be directly computed, as test labels are not observed.

## 4 METHODOLOGY

In this section, we build upon **TabPFN** [1] (Sec. 4.1), which performs ICL by conditioning on the training dataset but suffers from label shift. To address this, we propose **DistPFN** (Sec. 4.2), which reweights predicted class probabilities using the ratio between the posterior and the prior. We further introduce **DistPFN-T** (Sec. 4.3), which adaptively controls the adjustment strength using temperature scaling. Lastly, we propose a **inverse-frequency-based oversampling** method (Sec. 4.4) that oversamples rare classes in the training dataset based on inverse frequency to enable controlled evaluation under label shift. The overall framework of our method is described in Figure 3 and 4.

---

[1]We use **TabPFN** to refer broadly to the *family of ICL-based tabular foundation models*, including variants such as TabPFN-v2 (Hollmann et al., 2025), TabICL (Qu et al., 2025), and LoCalPFN (Thomas et al., 2024)

Figure 4: **TabPFN vs. DistPFN vs. DistPFN-T. (a)** As TabPFN suffers from majority-class bias under label shift, DistPFN mitigates this by adjusting its posterior distribution based on its own predictions. DistPFN-T further refines the adjustment by dynamically scaling its strength based on the discrepancy between the prior and posterior distributions. **(b)** Under label shift, both DistPFN and DistPFN-T outperform TabPFN, with DistPFN-T showing the most consistent improvements.

## 4.1 TABPFN

In tabular in-context learning, TabPFN predicts the label of a test instance $x_j$ by conditioning on the entire training dataset $\mathcal{D}_{\text{train}}$. Let $f(x_j, \mathcal{D}_{\text{train}}) \in \mathbb{R}^C$ denote the model output logits for the $C$ classes. The posterior distribution of TabPFN is computed via softmax:

$$\widehat{p}_{\text{TabPFN}}(y \mid x_j, \mathcal{D}_{\text{train}}) = \frac{\exp\left(f(x_j, \mathcal{D}_{\text{train}})[y]\right)}{\sum_{c=1}^{C} \exp\left(f(x_j, \mathcal{D}_{\text{train}})[c]\right)}, \tag{1}$$

where $[\cdot]$ denotes indexing over class logits. For simplicity, we denote the posterior distribution of any model $\widehat{p}(y \mid x_j, \mathcal{D}_{\text{train}})$ as $\widehat{p}(y)$, omitting the notations of input and training dataset.

## 4.2 DISTPFN: TEST-TIME POSTERIOR ADJUSTMENT

To improve the robustness under label shift, DistPFN extends TabPFN by adjusting the model's output, or the predicted distribution of $x_j$. Specifically, it introduces an adjustment factor based on the ratio between the predicted distribution $\widehat{p}_{\text{TabPFN}}(y)$ and the training prior $p_{\text{train}}(y)$ as:

$$\widetilde{p}_{\text{DistPFN}}(y) = \text{Norm}(\widehat{p}_{\text{TabPFN}}(y) \cdot \underbrace{\frac{\widehat{p}_{\text{TabPFN}}(y)}{p_{\text{train}}(y)}}_{\text{Adjustment factor } (\alpha)}) = \text{Norm}(\frac{\widehat{p}_{\text{TabPFN}}(y)^2}{p_{\text{train}}(y)}), \tag{2}$$

where $\text{Norm}(\cdot)$ denotes normalization over classes to ensure the probabilities sum to one. This adjustment down-weights the influence of the training prior and instead emphasizes the model's own prediction at test time, without requiring any modification to the model architecture or parameters.

Note that unlike conventional methods where the training distribution is only *indirectly* encoded in model parameters, ICL-based TabPFN *directly* conditions on the training dataset (i.e., context) at inference time, thereby enabling access to training data during prediction. This property makes our method specifically tailored to mitigate such bias in tabular foundation models.

## 4.3 DISTPFN-T: TEMPERATURE-SCALED ADJUSTMENT

While DistPFN corrects for label shift using the training prior, the optimal strength of adjustment may vary depending on *the deviation of test-time predictions from the training prior*. To make posterior adjustment more adaptive to the discrepancy between the training and the test dataset, we propose DistPFN-T, which introduces temperature scaling to control the sharpness of adjustment based on the discrepancy between the training prior $p_{\text{train}}(y)$ and the predicted distribution $\widehat{p}_{\text{TabPFN}}(y)$. Specifically, a temperature value $\tau$ is computed using the cross-entropy (CE) between the two distributions, where the predicted distribution $\widehat{p}_{\text{TabPFN}}(y)$ is then passed through a temperature-scaled softmax as:

$$\widehat{p}_{\text{TabPFN-T}}(y = c) = \frac{\exp\left(\widehat{p}_{\text{TabPFN}}(y = c)/\tau\right)}{\sum_{c'=1}^{C} \exp\left(\widehat{p}_{\text{TabPFN}}(y = c')/\tau\right)}, \quad \text{where} \quad \tau = \text{CE}(\widehat{p}_{\text{TabPFN}}(y), p_{\text{train}}(y)). \tag{3}$$

When the predicted distribution strongly deviates from the training prior (i.e., high cross-entropy), a high temperature is applied to smooth the predictions, thereby preventing over-adjustment. This temperature-scaled distribution $\widehat{p}_{\text{TabPFN-T}}(y)$ is used as the numerator of the adjustment factor in DistPFN-T, replacing $\widehat{p}_{\text{TabPFN}}(y)$ used in DistPFN as:

$$\widetilde{p}_{\text{DistPFN-T}}(y) = \text{Norm}(\widehat{p}_{\text{TabPFN}}(y) \cdot \underbrace{\frac{\widehat{p}_{\text{TabPFN-T}}(y)}{p_{\text{train}}(y)}}_{\text{Adjustment factor } (\alpha)}). \tag{4}$$

**Algorithm 1** Pseudocode for DistPFN

```
# x_test: test instance(s)
# D_train: training dataset
# p_train: training class prior
# f: TabPFN-based model
# alpha: adjustment factor
# method: ["tabpfn","distpfn","distpfn-t"]

logits = f(x_test, D_train)
p_hat = softmax(logits, dim=0)

if method == "tabpfn":
    alpha = 1

elif method == "distpfn":
    alpha = p_hat/p_train

elif method == "distpfn-t":
    tau = cross_entropy(p_hat, p_train)
    p_hat_scaled = softmax(p_hat/tau, dim=0)
    alpha = p_hat_scaled/p_train

p_hat = alpha * p_hat
return p_hat/p_hat.sum(dim=0)
```

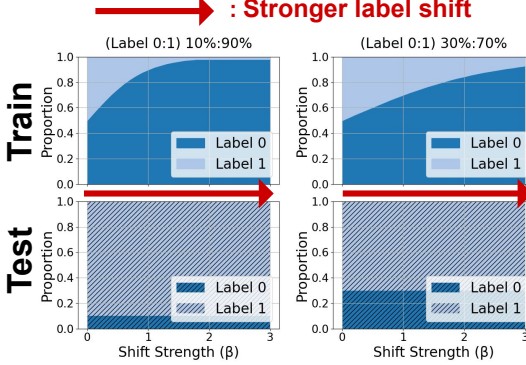

Figure 5: **Inverse-frequency-based oversampling.** As $\beta$ increases, the training distribution becomes increasingly biased toward rare classes, while the test distribution remains fixed, enabling fair comparison across varying degrees of shift.

Table 2 shows the example where DistPFN-T smooths the adjusted probabilities of DistPFN based on the prediction of TabPFN and the training prior. When the prediction of TabPFN aligns with the majority class (i.e., leans toward the majority class in the training prior), as in **Case 1) Majority**, DistPFN-T amplifies the *minority* class more than DistPFN, potentially inducing over-prediction. Conversely, when the prediction of

| Prediction | Case 1) **Majority** | | Case 2) **Minority** | |
|---|---|---|---|---|
| Class | A | B | A | B |
| Prior | 0.80 | 0.20 | 0.80 | 0.20 |
| TabPFN | **0.60** | 0.40 (-) | 0.40 | **0.60** (-) |
| + DistPFN | 0.36 | **0.64** (↑) | 0.10 | **0.90** (⇈) |
| + DistPFN-T | 0.33 | **0.67** (⇈) | 0.12 | **0.88** (↑) |

Table 2: Prediction w/ DistPFN and DistPFN-T.

TabPFN aligns with the minority class, as in **Case 2) Minority**, DistPFN-T amplifies the *majority* prediction to mitigate overprediction. This design is intuitive, as it counterbalances the bias introduced by the label distribution and the model's own prediction, leading to more calibrated and stable outputs. The effectiveness of DistPFN-T over DistPFN is demonstrated in Table 3 and Figure 7, using three different foundation models under varying degrees of distribution shift.

As TabPFN takes the entire test dataset (i.e., multiple instances) as input at once rather than processing each instance individually (i.e., single instance), the proposed methods apply their adjustment based on either the predicted distribution of a single instance or the average distribution of the test dataset, with the latter used by default in our experiments. A comparison of these two strategies is presented in Table 9, demonstrating the robustness of this design choice.

### 4.4 BENCHMARK FOR LABEL SHIFT: INVERSE-FREQUENCY-BASED OVERSAMPLING

To enable controlled evaluation under label shift, we propose *inverse-frequency-based oversampling*, which modifies the label distribution of $\mathcal{D}_{\text{train}}$ by oversampling each class according to its inverse frequency, while keeping $\mathcal{D}_{\text{test}}$ unchanged for direct comparison with the non-shifted setting. Note that we adopt oversampling rather than undersampling to avoid potential performance degradation caused by the removal of training instances. We define the class frequency in a dataset $\mathcal{D}$ consisting of $N$ samples as $p(y = c_k) = \frac{|i \in [N] : y_i = c_k|}{N}$, where the class-wise sampling weights are computed using their inverse frequency, assigning higher weights to rarer classes in the original distribution as:

$$w_k = \left(\frac{1}{p(y = c_k)}\right)^{\beta}, \qquad \tilde{w}_k = \frac{w_k}{\sum_{j=1}^{C} w_j},$$

with $\beta \geq 0$ controlling the strength of the shift.

Figure 5 illustrates the class distributions of the training and test datasets after oversampling, with respect to 1) the shift strength ($\beta$) and 2) the class distribution of the entire dataset. Higher values of $\beta$ assign higher sampling probabilities to rare classes, inducing a stronger shift, whereas $\beta = 0$ corresponds to uniform sampling (i.e., equal proportions across all classes). Note that $\beta = 0$ does not imply the absence of label shift, as the dataset itself may be class-imbalanced.

| Methods | | w/o shift | Shift strength ($\beta$) | | | | | | Avg. |
|---|---|---|---|---|---|---|---|---|---|
| | | | 0.0 | 0.1 | 0.5 | 1.0 | 2.0 | 5.0 | |
| Machine Learning | LogReg. | $0.765_{\pm0.002}$ | $0.719_{\pm0.002}$ | $0.709_{\pm0.002}$ | $0.674_{\pm0.003}$ | $0.634_{\pm0.004}$ | $0.597_{\pm0.002}$ | $0.566_{\pm0.004}$ | 0.650 |
| | + HPO | $0.771_{\pm0.003}$ | $0.697_{\pm0.002}$ | $0.687_{\pm0.003}$ | $0.653_{\pm0.004}$ | $0.616_{\pm0.004}$ | $0.586_{\pm0.003}$ | $0.550_{\pm0.003}$ | 0.631 |
| | SVM | $0.780_{\pm0.003}$ | $0.684_{\pm0.002}$ | $0.646_{\pm0.004}$ | $0.560_{\pm0.005}$ | $0.531_{\pm0.003}$ | $0.486_{\pm0.004}$ | $0.448_{\pm0.004}$ | 0.559 |
| | + HPO | $0.784_{\pm0.003}$ | $0.731_{\pm0.001}$ | $0.689_{\pm0.008}$ | $0.626_{\pm0.003}$ | $0.597_{\pm0.005}$ | $0.570_{\pm0.005}$ | $0.541_{\pm0.007}$ | 0.626 |
| | MLP | $0.778_{\pm0.004}$ | $0.658_{\pm0.005}$ | $0.647_{\pm0.004}$ | $0.613_{\pm0.005}$ | $0.574_{\pm0.006}$ | $0.527_{\pm0.004}$ | $0.493_{\pm0.001}$ | 0.585 |
| | + HPO | $0.795_{\pm0.005}$ | $0.706_{\pm0.006}$ | $0.688_{\pm0.006}$ | $0.654_{\pm0.008}$ | $0.615_{\pm0.006}$ | $0.580_{\pm0.005}$ | $0.545_{\pm0.005}$ | 0.631 |
| | $k$NN | $0.765_{\pm0.004}$ | $0.663_{\pm0.004}$ | $0.657_{\pm0.004}$ | $0.629_{\pm0.003}$ | $0.589_{\pm0.003}$ | $0.538_{\pm0.003}$ | $0.501_{\pm0.004}$ | 0.596 |
| | + HPO | $0.783_{\pm0.002}$ | $0.693_{\pm0.002}$ | $0.684_{\pm0.003}$ | $0.644_{\pm0.004}$ | $0.588_{\pm0.003}$ | $0.540_{\pm0.003}$ | $0.498_{\pm0.002}$ | 0.608 |
| | Random Forest | $0.796_{\pm0.003}$ | $0.768_{\pm0.003}$ | $0.765_{\pm0.003}$ | $0.748_{\pm0.005}$ | $0.718_{\pm0.004}$ | $0.665_{\pm0.005}$ | $0.618_{\pm0.005}$ | 0.714 |
| | + HPO | $0.803_{\pm0.002}$ | $0.771_{\pm0.002}$ | $0.767_{\pm0.001}$ | $0.743_{\pm0.004}$ | $0.701_{\pm0.004}$ | $0.627_{\pm0.008}$ | $0.578_{\pm0.006}$ | 0.698 |
| | LightGBM | $0.789_{\pm0.003}$ | $0.758_{\pm0.004}$ | $0.753_{\pm0.002}$ | $0.734_{\pm0.004}$ | $0.705_{\pm0.004}$ | $0.657_{\pm0.005}$ | $0.618_{\pm0.005}$ | 0.704 |
| | + HPO | $0.790_{\pm0.006}$ | $0.726_{\pm0.008}$ | $0.661_{\pm0.005}$ | $0.655_{\pm0.008}$ | $0.608_{\pm0.008}$ | $0.577_{\pm0.015}$ | $0.551_{\pm0.004}$ | 0.630 |
| | CatBoost | $0.803_{\pm0.001}$ | $0.774_{\pm0.002}$ | $0.771_{\pm0.002}$ | $0.751_{\pm0.004}$ | $0.718_{\pm0.004}$ | $0.665_{\pm0.005}$ | $0.621_{\pm0.005}$ | 0.717 |
| | + HPO | $0.802_{\pm0.002}$ | $0.774_{\pm0.002}$ | $0.771_{\pm0.002}$ | $0.752_{\pm0.004}$ | $0.719_{\pm0.004}$ | $0.665_{\pm0.006}$ | $0.621_{\pm0.005}$ | 0.717 |
| Deep Learning — Non-found. | FT-Transformer | $0.784_{\pm0.002}$ | $0.748_{\pm0.004}$ | $0.746_{\pm0.004}$ | $0.718_{\pm0.005}$ | $0.674_{\pm0.005}$ | $0.610_{\pm0.003}$ | $0.551_{\pm0.007}$ | 0.675 |
| | TabM | $0.794_{\pm0.002}$ | $0.762_{\pm0.004}$ | $0.757_{\pm0.004}$ | $0.735_{\pm0.003}$ | $0.694_{\pm0.005}$ | $0.624_{\pm0.006}$ | $0.565_{\pm0.006}$ | 0.690 |
| | TabulaRNN | $0.749_{\pm0.003}$ | $0.699_{\pm0.004}$ | $0.684_{\pm0.004}$ | $0.641_{\pm0.004}$ | $0.585_{\pm0.009}$ | $0.522_{\pm0.011}$ | $0.465_{\pm0.008}$ | 0.599 |
| | MambaTab | $0.719_{\pm0.004}$ | $0.629_{\pm0.006}$ | $0.603_{\pm0.004}$ | $0.525_{\pm0.002}$ | $0.466_{\pm0.010}$ | $0.430_{\pm0.002}$ | $0.394_{\pm0.002}$ | 0.508 |
| | RealMLP | $0.794_{\pm0.002}$ | $0.760_{\pm0.004}$ | $0.758_{\pm0.005}$ | $0.745_{\pm0.003}$ | $0.720_{\pm0.005}$ | $0.677_{\pm0.002}$ | $0.643_{\pm0.004}$ | 0.717 |
| Deep Learning — Foundation | LoCalPFN | $\mathbf{0.816}_{\pm0.002}$ | $0.794_{\pm0.003}$ | $0.793_{\pm0.004}$ | $0.788_{\pm0.003}$ | $0.778_{\pm0.002}$ | $0.753_{\pm0.004}$ | $0.719_{\pm0.000}$ | 0.771 |
| | + DistPFN | $\mathbf{0.816}_{\pm0.002}$ | $0.797_{\pm0.001}$ | $0.796_{\pm0.002}$ | $0.794_{\pm0.002}$ | $0.790_{\pm0.002}$ | $0.782_{\pm0.001}$ | $0.770_{\pm0.003}$ | 0.788 |
| | + DistPFN-T | $\mathbf{0.816}_{\pm0.002}$ | $0.798_{\pm0.002}$ | $0.797_{\pm0.002}$ | $0.796_{\pm0.002}$ | $0.794_{\pm0.002}$ | $0.787_{\pm0.001}$ | $0.776_{\pm0.003}$ | 0.791 |
| | TabICL | $\mathbf{0.806}_{\pm0.002}$ | $0.783_{\pm0.002}$ | $0.781_{\pm0.002}$ | $0.770_{\pm0.002}$ | $0.747_{\pm0.002}$ | $0.704_{\pm0.006}$ | $0.664_{\pm0.006}$ | 0.742 |
| | + DistPFN | $\mathbf{0.806}_{\pm0.002}$ | $0.786_{\pm0.002}$ | $0.786_{\pm0.002}$ | $0.781_{\pm0.002}$ | $0.776_{\pm0.002}$ | $0.763_{\pm0.002}$ | $0.746_{\pm0.004}$ | 0.773 |
| | + DistPFN-T | $\mathbf{0.806}_{\pm0.003}$ | $0.786_{\pm0.003}$ | $0.786_{\pm0.003}$ | $0.783_{\pm0.002}$ | $0.780_{\pm0.002}$ | $0.771_{\pm0.001}$ | $0.755_{\pm0.004}$ | 0.777 |
| | TabPFN-v2 | $\mathbf{0.818}_{\pm0.004}$ | $0.797_{\pm0.003}$ | $0.796_{\pm0.004}$ | $0.790_{\pm0.002}$ | $0.782_{\pm0.002}$ | $0.759_{\pm0.003}$ | $0.727_{\pm0.003}$ | 0.775 |
| | + DistPFN | $\mathbf{0.818}_{\pm0.002}$ | $0.799_{\pm0.001}$ | $0.797_{\pm0.002}$ | $0.795_{\pm0.002}$ | $0.791_{\pm0.003}$ | $0.783_{\pm0.003}$ | $0.769_{\pm0.003}$ | 0.789 |
| | + DistPFN-T | $\mathbf{0.818}_{\pm0.002}$ | $0.799_{\pm0.003}$ | $0.798_{\pm0.002}$ | $0.797_{\pm0.002}$ | $0.796_{\pm0.003}$ | $0.789_{\pm0.003}$ | $0.775_{\pm0.003}$ | 0.792 |

Table 3: **Tabular classification results.** While most baselines suffer substantial performance degradation under label shift, our methods significantly improve the accuracy of ICL-based tabular foundation models (e.g., TabPFN-v2) across varying degrees of shift ($\beta$), averaged over 253 datasets.

# 5 EXPERIMENTS

## 5.1 EXPERIMENTAL SETUP

**Task and metrics.** We evaluate our methods on tabular classification tasks both with and without label shift to assess their robustness to label shift. For the evaluation metrics, we employ accuracy (Acc.), ROC-AUC, and average rank (Rank), following the previous works (Hollmann et al., 2023). Further details regarding the experimental setups are provided in Appendix A.

**Datasets.** We evaluate our methods on 253 tabular classification datasets from OpenML (Bischl et al., 2017), which span a wide range of feature dimensions, class cardinalities, sample sizes, and domains. Unless otherwise specified, performance is reported as the mean accuracy across all datasets, averaged over five different random seeds. Additionally, to specifically assess performance under label shift, we construct synthetic variants by modifying the test set while keeping the training set fixed, following the standard setup for fair comparison, as described in Section 4.4. Following the previous works (Hollmann et al., 2023; 2025), all methods are evaluated using the fixed train/test splits, where each dataset is randomly split into 50% training and 50% test data.

**Baseline models.** We categorize a total of 15 baseline tabular models into the following three groups:

- **ML models (7):** Logistic Regression (LR), Support Vector Machines (SVM), Random Forest (Liaw & Wiener, 2002), $k$-nearest neighbors ($k$NN), Multi-layer Perceptrons (MLP), LightGBM (Ke et al., 2017), CatBoost (Prokhorenkova et al., 2018)
- **DL (non-foundation) models (5):** FT-Transformer (Gorishniy et al., 2021), TabM (Gorishniy et al., 2024a), TabulaRNN (Thielmann & Samiee, 2024), MambaTab (Ahamed & Cheng, 2024), RealMLP (Holzmüller et al., 2024)
- **DL (foundation) models based on ICL (3):** TabPFN-v2 (Hollmann et al., 2025), LoCalPFN (Thomas et al., 2024), TabICL (Qu et al., 2025)

Additionally, we perform hyperparameter optimization (HPO) for ML models[2] for stronger baselines, using the search space provided in a public implementation, with details provided in Appendix F.

---

[2]While MLP can be regarded as a DL model, we categorize it as an ML model for the purpose of HPO.

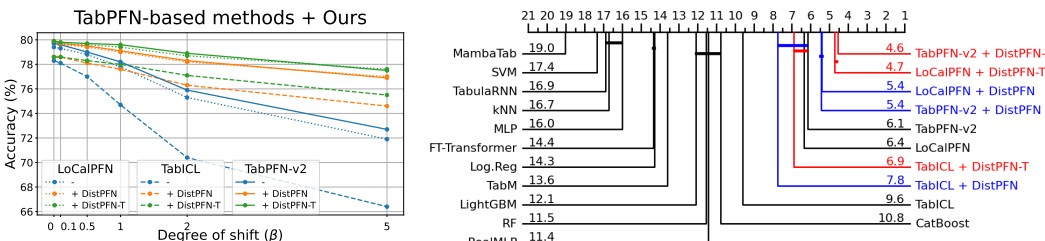

Figure 7: Performance by shift.

Figure 8: Rank (CD Diagram) under shift ($\beta = 2$).

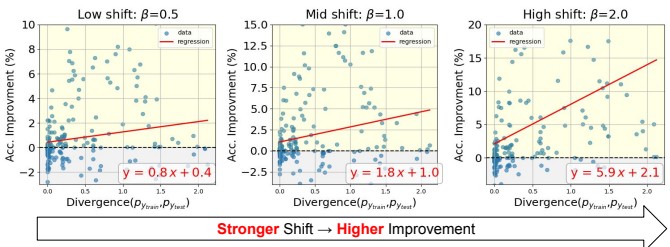

Figure 9: **Per-dataset improvement.** The figure shows the accuracy improvement for each dataset under varying $\beta$ values with DistPFN-T applied, shown against the KL-divergence between the train and test label distributions of the original dataset.

Figure 10: **vs. Oracle.** The figure compares the performance of our method to DistPFN-Oracle, which uses the ground-truth test label distribution as the adjustment factor.

## 5.2 CLASS-IMBALANCED BENCHMARK DATASETS

We examine the degree of class imbalance across 253 OpenML datasets by defining the *balance ratio* as the number of samples in the minority class ($N_{\text{minority}}$) divided by the number of samples in the majority class ($N_{\text{majority}}$). A balance ratio of 100% corresponds to a perfectly balanced dataset, while lower values indicate increasing imbalance. As shown in Figure 6, approximately 85% of the datasets exhibit class imbalance, highlighting the importance of addressing the majority-class bias.

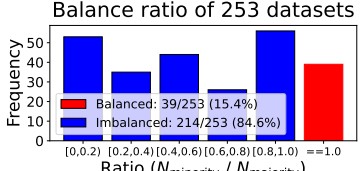

Figure 6: Balance ratio of datasets.

## 5.3 TABULAR CLASSIFICATION

Table 3 reports the average accuracy over 253 datasets across six levels of label shift ($\beta$), comparing our method against 16 baselines, including three tabular foundation models to which our method is applied. For LoCalPFN, we use the $k = 10$ nearest neighbors for each test sample, with robustness to $k$ demonstrated in Table 5. While maintaining the original performance under the standard setting without label shift, our method substantially improves all three foundation models under shift, without any parameter update or additional computational cost, as shown in Table 11.

Figure 7 shows that as label shift strength $\beta$ increases, our methods improve all three foundation models, yielding larger gains at higher $\beta$, with DistPFN-T providing additional improvements over DistPFN. For instance, DistPFN and DistPFN-T improve the accuracy of TabICL by 12.3% and 13.7% under $\beta = 5$, respectively. Additionally, Figure 8 shows the average rank across datasets under $\beta = 2$ using a critical difference (CD) diagram, with DistPFN-T applied to TabPFN-v2 achieving SoTA performance. A comparison based on ROC-AUC is provided in Appendix L.

## 6 ANALYSIS

In this section, we conduct various analyses on the effectiveness of our methods, DistPFN and DistPFN-T, which are applied to TabPFN-v2 (Hollmann et al., 2025) unless otherwise stated.

**Performance gain of each dataset.** Figure 9 illustrates the accuracy improvement across 253 datasets under three different $\beta$ values when DistPFN-T is applied. Each point represents a dataset, with the x-axis showing the KL-divergence between the train and test label distributions of the original dataset, and the y-axis indicating the corresponding accuracy improvement. As $\beta$ increases, datasets with larger original discrepancies become more imbalanced, with those exhibiting stronger divergence (i.e., larger induced shifts) benefiting more from our method.

| $\alpha = \dfrac{①}{p_{\text{train}}(y)}$ | ① | w/o shift | Shift strength ($\beta$) | | | | | | |
|---|---|---|---|---|---|---|---|---|---|
| | | | 0.0 | 0.1 | 0.5 | 1.0 | 2.0 | 5.0 | Avg. |
| TabPFN-v2 | - | **0.818** | 0.797 | 0.796 | 0.790 | 0.782 | 0.759 | 0.727 | 0.775 |
| + DistPFN | $\widehat{p}_{\text{TabPFN}}(y)$ | **0.818** | 0.799 | 0.797 | 0.795 | 0.791 | 0.783 | 0.769 | 0.789 |
| + DistPFN-T | $\widehat{p}_{\text{DistPFN-T}}(y)$ | **0.818** | 0.799 | 0.798 | 0.797 | 0.796 | 0.789 | 0.775 | 0.792 |
| + DistPFN-Oracle | $p_{\text{test}}(y)$ | **0.818** | **0.803** | **0.802** | **0.800** | **0.797** | **0.792** | **0.784** | **0.796** |

Table 4: **Comparison with oracle.** While our method computes the adjustment factor $\alpha$ using the *predicted* test label distribution, we also compare it to *DistPFN-Oracle*, which instead uses the *true* test label ratio. The results demonstrate that our methods achieve performance close to this oracle.

| $k$ | Methods | Avg. |
|---|---|---|
| 3 | LoCalPFN | 0.750 |
| | + DistPFN | 0.782 |
| | + DistPFN-T | **0.785** |
| 10 | LoCalPFN | 0.770 |
| | + DistPFN | 0.787 |
| | + DistPFN-T | **0.789** |
| 20 | LoCalPFN | 0.771 |
| | + DistPFN | 0.788 |
| | + DistPFN-T | **0.791** |

Table 5: Application to LoCalPFN.

| $\alpha = \dfrac{\widehat{p}_{\text{DistPFN(-T)}}(y)}{②}$ | ② | w/o shift | Shift strength ($\beta$) | | | | | | |
|---|---|---|---|---|---|---|---|---|---|
| | | | 0.0 | 0.1 | 0.5 | 1.0 | 2.0 | 5.0 | Avg. |
| TabPFN-v2 | - | 0.818 | 0.797 | 0.796 | 0.790 | 0.782 | 0.759 | 0.727 | 0.775 |
| + DistPFN | $p_{\text{train}}(y)$ | **0.818** | **0.799** | **0.797** | **0.795** | 0.791 | **0.783** | **0.769** | **0.789** |
| | $\widehat{p}_{\text{train}}(y)$ | **0.818** | **0.799** | **0.797** | **0.795** | **0.792** | **0.783** | 0.768 | **0.789** |
| + DistPFN-T | $p_{\text{train}}(y)$ | **0.818** | 0.799 | 0.798 | 0.797 | 0.796 | 0.789 | 0.775 | 0.792 |
| | $\widehat{p}_{\text{train}}(y)$ | **0.818** | **0.800** | **0.800** | **0.798** | **0.797** | **0.791** | **0.777** | **0.793** |

Table 6: **Training prior vs. Training prediction.** Replacing the training prior $p_{\text{train}}(y)$ with the predicted distribution $\widehat{p}_{\text{train}}(y)$ in $\alpha$ shows negligible performance difference, validating the use of $p_{\text{train}}(y)$ as a simple and reliable choice, as $\widehat{p}_{\text{train}}(y)$ requires additional computation.

| ①: $p_{\text{train}}$, ②: $\widehat{p}_{\text{TabPFN}}(y)$ | | w/o shift | Shift strength ($\beta$) | | | | | | |
|---|---|---|---|---|---|---|---|---|---|
| | | | 0.0 | 0.1 | 0.5 | 1.0 | 2.0 | 5.0 | Avg. |
| TabPFN-v2 | | **0.818** | 0.797 | 0.796 | 0.790 | 0.782 | 0.759 | 0.727 | 0.775 |
| + DistPFN-T | $\tau = \text{CE}(①, ②)$ | **0.818** | **0.799** | **0.799** | **0.797** | 0.795 | 0.788 | 0.769 | 0.791 |
| | $\tau = \text{CE}(②, ①)$ | **0.818** | **0.799** | 0.798 | **0.797** | **0.796** | **0.789** | **0.775** | **0.792** |

Table 7: **Asymmetric cross-entropy.** DistPFN-T employs asymmetric cross-entropy to compute the temperature $\tau$ for temperature scaling, where both directions outperform TabPFN-v2.

**Comparison with oracle.** The adjustment factor $\alpha$ in our method is computed using the *predicted* label distribution. We compare this to a variant that uses the *true* label ratio of the test set, referred to as *DistPFN-Oracle*, which is unavailable in practice. This replaces the *predicted* distribution with the *true* distribution as the numerator of $\alpha$. Table 4 and Figure 10 show the results, indicating that our methods achieve performance close to the oracle even without access to the (ground truth) test label.

**Robustness to $k$ for LoCalPFN.** We apply our methods to LoCalPFN (Thomas et al., 2024), which improves the efficiency of TabPFN by retrieving $k$ nearest neighbors for each test sample to construct the training dataset. For a fair comparison, we do not fine-tune the model and instead use the pretrained weights of TabPFN-v2 (Hollmann et al., 2025), providing a stronger baseline than the original TabPFN (Hollmann et al., 2023) used in LoCalPFN. Table 5 reports results across different values of number of neighbors ($k$), averaged over six $\beta$ values. The results indicate that our methods consistently improve performance, with full results provided in Appendix I.

**Training prior vs. Training prediction.** The adjustment factor $\alpha$ of our method uses the ground-truth label distribution of the training dataset (i.e., *training prior* or $p_{\text{train}}(y)$) as the denominator, while the numerator is based on the predicted distribution of the test set, introducing a mismatch between true and predicted quantities. To assess the impact of this discrepancy, we analyze an alternative that replaces the training prior with the model's average predicted distribution on the training dataset (i.e., *training prediction* or $\widehat{p}_{\text{train}}(y)$), which requires additional inference. As shown in Table 6, the performance difference is negligible, validating the choice of the training prior.

**Asymmetric cross-entropy (CE).** DistPFN-T employs asymmetric cross-entropy to compute the temperature $\tau$ for temperature scaling, where the value differs depending on whether it is computed as $\text{CE}(\widehat{p}_{\text{TabPFN}}(y), p_{\text{train}})$ or $\text{CE}(p_{\text{train}}, \widehat{p}_{\text{TabPFN}}(y))$. Table 7 shows that both directions outperform TabPFN-v2, demonstrating robustness to the choice of direction.

| Methods | LoCalPFN | | TabICL | | TabPFN-v2 | |
|---|---|---|---|---|---|---|
| | w/o shift | w/ shift | w/o shift | w/ shift | w/o shift | w/ shift |
| - | **0.816** | 0.771 | **0.806** | 0.742 | **0.818** | 0.775 |
| + EME | 0.801 | 0.783 | 0.798 | 0.766 | 0.801 | 0.786 |
| + BBE | 0.805 | 0.787 | 0.802 | 0.768 | 0.805 | 0.789 |
| + DistPFN | **0.816** | 0.788 | **0.806** | 0.773 | **0.818** | 0.789 |
| + DistPFN-T | **0.816** | **0.791** | **0.806** | **0.777** | **0.818** | **0.792** |

Table 8: Comparison with methods for label shift.

| | Pred. distn. | w/o shift | w/ shift |
|---|---|---|---|
| TabPFN-v2 | - | | 0.818 | 0.775 |
| + DistPFN | Single | **0.818** | **0.789** |
| | Multiple | **0.818** | **0.789** |
| + DistPFN-T | Single | **0.818** | 0.791 |
| | Multiple | **0.818** | **0.792** |

Table 9: Pred distn: single vs. multiple.

| P | Methods | Shift strength ($\beta$) | | | | | | |
|---|---|---|---|---|---|---|---|---|
| | | 0.0 | 0.1 | 0.5 | 1.0 | 2.0 | 5.0 | Avg. |
| 0.05 | TabPFN-v2 | 0.644 | 0.639 | 0.591 | 0.547 | 0.505 | 0.460 | 0.554 |
| | + DistPFN | 0.659 | 0.662 | 0.627 | 0.586 | 0.548 | 0.493 | 0.589 |
| | + DistPFN-T | **0.662** | **0.667** | **0.640** | **0.601** | **0.562** | **0.504** | **0.605** |
| 0.10 | TabPFN-v2 | 0.663 | 0.664 | 0.617 | 0.582 | 0.534 | 0.481 | 0.620 |
| | + DistPFN | 0.679 | 0.685 | 0.650 | 0.618 | 0.577 | 0.515 | 0.642 |
| | + DistPFN-T | **0.685** | **0.691** | **0.656** | **0.629** | **0.588** | **0.527** | **0.652** |
| 0.20 | TabPFN-v2 | 0.697 | 0.689 | 0.651 | 0.619 | 0.561 | 0.510 | 0.638 |
| | + DistPFN | 0.713 | 0.711 | 0.682 | 0.663 | 0.610 | 0.553 | 0.676 |
| | + DistPFN-T | **0.716** | **0.715** | **0.689** | **0.670** | **0.620** | **0.563** | **0.688** |

Table 10: **Dataset selection with K-means clustering.** Our method remains effective when training subsets are formed by sampling a proportion ($P$) from each of the $K = 10$ clusters, demonstrating robustness to the choice of training subsets.

| | Pred. time | Avg. Acc. |
|---|---|---|
| LoCalPFN | 0.618 | 0.771 |
| + DistPFN | 0.619 | 0.788 |
| + DistPFN-T | 0.619 | **0.791** |
| TabICL | 0.620 | 0.742 |
| + DistPFN | 0.622 | 0.773 |
| + DistPFN-T | 0.622 | **0.777** |
| TabPFN-v2 | 1.002 | 0.775 |
| + DistPFN | 1.003 | 0.789 |
| + DistPFN-T | 1.003 | **0.792** |

Table 11: **Efficiency analysis.** Average prediction time (in seconds) average across 253 datasets with three different backbones.

**Comparison label shift correction methods.** To demonstrate the effectiveness of our methods, we compare it with other techniques handling label shift: EM-based Estimation (**EME**) (Saerens et al., 2002) and Black-box Estimation (**BBE**) (Lipton et al., 2018). Table 8 demonstrates that our method outperforms these approaches without requiring estimation of the test prior. In particular, as shown in Figure 11, while other methods suffer from performance degradation when no shift is present, our method maintains stable performance across both settings. Details of each method and full results are provided in Appendix E and J.

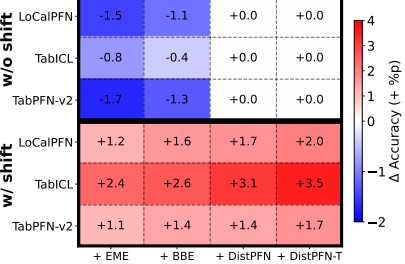

Figure 11: vs. Other label shift methods.

**Predicted distribution of single vs. multiple instance(s).** As TabPFN allows test instances to be evaluated either individually or in batches, with both modes yielding identical predictions, DistPFN and DistPFN-T can apply their adjustment based on either the 1) prediction of a *single* instance or 2) the average prediction across *multiple* instances in the test dataset. As shown in Table 9, both choices consistently improve TabPFN-v2, averaged across six $\beta$s for *w/ shift*. The results demonstrate that our method is robust to the choice of distribution source, with full results shown in Appendix K.

**Training set selection.** TabPFN suffers from quadratic complexity, making it inefficient for large training datasets (Thomas et al., 2024; Zeng et al., 2025; Qu et al., 2025), and several works mitigate this by training on *selected* subsets. One common approach is to use only the local neighbors of each test sample, as validated in Table 3 with LoCalPFN (Thomas et al., 2024). Another approach clusters the training dataset and selects the centroid and a few nearby samples per cluster. As shown in Table 10, our method remains effective across different percentages of samples per cluster ($P$) under $K = 10$, where $K$ is the number of clusters. Results for various $K$s are provided in Appendix H.

**Efficiency analysis.** To evaluate the efficiency of our method, we compare the average prediction time (seconds) across 253 datasets using TabPFN-v2. Note that our method does not require any additional parameters, and only applies a simple multiplication of an adjustment factor to the predicted results. Table 11 summarizes the results, including the average performance across six $\beta$s, highlighting that our method achieves superior performance gains with negligible computational burden.

# 7 CONCLUSION

In this work, we introduce DistPFN, a test-time adjustment method to mitigate label shift in tabular foundation models using ICL. We further propose DistPFN-T to stabilize the adjustment via temperature scaling based on distributional divergence between training prior and predicted distribution. While our method effectively handles label shift without retraining, it does not address feature shift, which can also occur in practice. A potential direction for future work is to design foundation models that are inherently robust to both label and feature shift, beyond post-hoc adjustment. We hope that this work encourages further exploration of robustness to distribution shifts in tabular ICL.

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

APPENDIX

## A    Experimental Setups

**Experimental setups.** We use the official implementation of TabPFN[3] and adopt all default settings without modification. This includes architectural choices such as the number of layers and hidden dimensions, where we use 12 transformer layers, each with a hidden size of 192 and 6 attention heads. The feedforward layer dimension is implicitly set to 768 via a hidden factor of 4. For inference, we load the pretrained weights from TabPFN-v2[4] available on Hugging Face.

**Dataset.** We evaluate on 250+ tabular datasets from OpenML (Bischl et al., 2017). The dataset list is retrieved from the benchmark configuration provided in this repository[5], which is built on top of the official TabPFN evaluation setup. Dataset statistics are summarized in Appendix G.

## B    Baseline Methods

We categorize 15 baseline tabular models into three groups:

- **ML models (7):** Logistic Regression (LR), Support Vector Machines (SVM), Random Forest (Liaw & Wiener, 2002), $k$-nearest neighbors ($k$NN), Multi-layer Perceptrons (MLP), LightGBM (Ke et al., 2017), CatBoost (Prokhorenkova et al., 2018)
- **DL (non-foundation) models (5):** FT-Transformer (Gorishniy et al., 2021), TabM (Gorishniy et al., 2024a), TabulaRNN (Thielmann & Samiee, 2024), MambaTab (Ahamed & Cheng, 2024), RealMLP (Holzmüller et al., 2024)
- **DL (foundation) models based on ICL (3):** TabPFN-v2 (Hollmann et al., 2025), LoCalPFN (Thomas et al., 2024), TabICL (Qu et al., 2025)

Details of each method are provided below.

### B.1    Machine Learning (ML) Models

- **Logistic Regression (LR)** (Cox, 1958): A simple linear model commonly used for binary and multiclass classification tasks in tabular data.
- **Support Vector Machine (SVM)** (Cortes & Vapnik, 1995): A kernel-based classifier that aims to find the optimal decision boundary with maximum margin between classes.
- **Multilayer Perceptron (MLP)** (Haykin, 1994): A feedforward neural network consisting of multiple fully connected layers with non-linear activations, trained via backpropagation.
- **$k$-Nearest Neighbors** ($k$NN) (Altman, 1992): A non-parametric method that classifies a sample based on the majority class among its $k$ nearest neighbors in the feature space.
- **Random Forest** (Liaw & Wiener, 2002): An ensemble learning method based on bagging over decision trees, which improves robustness and generalization.
- **LightGBM (Ke et al., 2017)**: A fast and efficient GBDT model using histogram-based algorithms and leaf-wise tree growth.
- **CatBoost (Prokhorenkova et al., 2018)**: A GBDT model that handles categorical features efficiently and mitigates prediction shift via ordered boosting.

---

[3]https://github.com/PriorLabs/TabPFN
[4]https://huggingface.co/Prior-Labs/TabPFN-v2-clf
[5]https://github.com/carteakey/tabpfn-eval/blob/main/src/data/openml_list.csv

## B.2 DEEP LEARNING (NON-FOUNDATION) MODELS

- **FT-Transformer (Gorishniy et al., 2021)**: A transformer-based architecture tailored for tabular data, providing a simple yet powerful baseline that outperforms many prior DL models on classification and regression tasks.
- **TabM (Gorishniy et al., 2024a)**: An MLP-based model that leverages an efficient ensemble mechanism to approximate deep ensembles, enabling multiple predictions per instance while maintaining computational efficiency.
- **TabulaRNN (Thielmann & Samiee, 2024)**: An RNN-inspired architecture for tabular data that emphasizes efficiency, addressing limitations of NLP-style models in terms of scalability and training cost.
- **MambaTab (Ahamed & Cheng, 2024)**: A scalable and efficient model built on structured state-space models (SSMs), capturing long-range dependencies with fewer parameters while maintaining strong predictive performance.
- **RealMLP (Holzmüller et al., 2024)**: An enhanced MLP variant with meta-tuned hyperparameters, achieving competitive accuracy–efficiency trade-offs compared to gradient boosting methods in tabular benchmarks.

## B.3 DEEP LEARNING (FOUNDATION) MODELS BASED ON ICL

- **LoCalPFN (Thomas et al., 2024)**: A lightweight PFN variant that reduces computational cost by leveraging local task priors and architectural simplifications.
- **TabICL (Qu et al., 2025)**: A two-stage model that first applies column attention to capture feature dependencies and then row attention to encode sample interactions.
- **TabPFN-v2** (Hollmann et al., 2025): A state-of-the-art foundation model for tabular classification that leverages a pretrained transformer for zero-shot prediction on small datasets.

## C BASELINE IMPLEMENTATIONS

The baseline results are obtained from the following publicly available repositories:

- [1] **TabPFN Evaluation** framework[6] was used to evaluate all ML models, as well as the foundation models *TabPFN* and *LoCalPFN*. Since *LoCalPFN* does not have an official implementation, we reimplemented it based on the TabPFN codebase.
- [2] **AutoGluon v1.4.0**[7] was used to benchmark *TabICL* and several non-foundation models such as *FT-Transformer*, *TabM*, and *RealMLP*.
- [3] **Mambular**[8] provided implementations for additional non-foundation models including *MambaTab*, *TabulaRNN*, *FT-Transformer*, and *TabM*.

For models implemented in both [2] and [3] (e.g., FT-Transformer and TabM), we use the [2] versions as they yield stronger performance for a stronger baseline.

---

[6]https://github.com/carteakey/tabpfn-eval
[7]https://auto.gluon.ai/
[8]https://github.com/OpenTabular/DeepTabular

# D    THEORETICAL JUSTIFICATION

We provide theoretical grounding for our posterior adjustment to clarify that DistPFN is not merely a heuristic trick, but a principled approximation derived from existing theory. We present two complementary perspectives: 1) connection to classical label shift correction as a plug-in reweighting (Section D.1) and 2) Bayesian view that replaces the mismatched prior with a self-consistent estimate from model predictions (Section D.2).

## D.1    RELATION TO LABEL SHIFT CORRECTION

The label shift setting assumes that the conditional distribution $p(x|y)$ remains invariant while the marginal priors differ:

$$p_{\text{train}}(y) \neq p_{\text{test}}(y), \quad p(x|y) \text{ is fixed.}$$

Under this assumption, the Bayes-optimal posterior is given by

$$p_{\text{test}}(y|x) \propto \frac{p_{\text{train}}(y|x)}{p_{\text{train}}(y)} \, p_{\text{test}}(y).$$

Classical approaches such as EM-based reweighting (Saerens et al., 2002; Lipton et al., 2018) estimate $p_{\text{test}}(y)$ explicitly by matching marginal predictions to unlabeled test data. DistPFN instead uses the predictive marginal $\hat{p}(y)$ obtained directly from the model, and constructs the adjustment factor

$$\alpha(y) = \frac{\hat{p}(y)}{p_{\text{train}}(y)}.$$

This yields the corrected posterior

$$\hat{p}(y|x) \propto \frac{p_{\text{train}}(y|x)}{p_{\text{train}}(y)} \, \hat{p}(y),$$

which can be seen as a plug-in realization of the classical correction rule, avoiding iterative estimation while remaining theoretically consistent with label shift correction.

## D.2    BAYESIAN INTERPRETATION

From a Bayesian perspective, TabPFN models the posterior under the training distribution:

$$p_{\text{train}}(y|x) \propto p(x|y) \, p_{\text{train}}(y).$$

At test time, however, the desired posterior is

$$p_{\text{test}}(y|x) \propto p(x|y) \, p_{\text{test}}(y).$$

The difference comes solely from the prior. DistPFN addresses this gap by substituting $p_{\text{train}}(y)$ with $\hat{p}(y)$, the average predictive distribution obtained on the test set:

$$\hat{p}_{\text{DistPFN}}(y|x) \propto \frac{p_{\text{train}}(y|x)}{p_{\text{train}}(y)} \, \hat{p}(y).$$

This interpretation shows that DistPFN is not an ad-hoc adjustment but a Bayesian posterior correction where the unknown test prior is approximated in a self-consistent manner from model outputs. The method therefore inherits a principled justification while retaining the efficiency of a simple, training-free plug-in procedure.

# E  CLASSICAL METHODS FOR LABEL SHIFT CORRECTION

In this section, we summarize three representative approaches for handling label shift. All of these methods directly adjust classifier outputs, but they differ in how the test prior $\pi_{\text{test}}$ is obtained.

## E.1  PRIOR-RATIO ADJUSTMENT

Prior-ratio Adjustment (Elkan, 2001) introduces a simple correction under changing class priors in the binary setting. The method assumes that the new prior $\pi_{\text{test}}$ is available from external knowledge or domain statistics. Given a posterior $p_{\text{train}}(1|x)$ trained under $\pi_{\text{train}}$, the corrected posterior is

$$p_{\text{test}}(1|x) \;=\; \frac{p_{\text{train}}(1|x) \cdot \frac{\pi_{\text{test}}(1)}{\pi_{\text{train}}(1)}}{p_{\text{train}}(1|x) \cdot \frac{\pi_{\text{test}}(1)}{\pi_{\text{train}}(1)} + (1 - p_{\text{train}}(1|x)) \cdot \frac{1-\pi_{\text{test}}(1)}{1-\pi_{\text{train}}(1)}}.$$

This approach directly modifies posterior probabilities by scaling them with prior ratios. The same principle naturally extends to the multiclass case by applying the ratio $\pi_{\text{test}}(y)/\pi_{\text{train}}(y)$ to each class posterior.

## E.2  EM-BASED ESTIMATION

EM-based Estimation (Saerens et al., 2002) proposes an iterative procedure to estimate unknown test priors when they are not directly given. At iteration $t$, the posterior is updated by

$$p^{(t+1)}(y|x) \;\propto\; p_{\text{train}}(y|x) \cdot \frac{\pi_{\text{test}}^{(t)}(y)}{\pi_{\text{train}}(y)}.$$

The updated posteriors provide a new estimate of $\pi_{\text{test}}$ by averaging across the test set. Repeating this E-step and M-step allows the estimated test prior to gradually converge. The final corrected posterior then follows the standard prior-ratio adjustment, but with $\pi_{\text{test}}$ estimated rather than assumed.

## E.3  BLACK-BOX ESTIMATION

Black-box Estimation (Lipton et al., 2018) employs a validation dataset with true labels to construct a confusion matrix $C(s|y) = P(\hat{y} = s \mid y)$ that characterizes prediction errors of the classifier. On an unlabeled test set, it collects predicted labels to obtain the empirical distribution $p_{\text{test}}(s)$. These quantities are related through the equation

$$p_{\text{test}}(s) \;\approx\; \sum_{y} C(s|y)\,\pi_{\text{test}}(y).$$

By solving this linear system, the method estimates the test prior $\pi_{\text{test}}$. Once the test prior is recovered, the posterior correction is applied using the prior ratio:

$$p_{\text{test}}(y|x) \;\propto\; p_{\text{train}}(y|x) \cdot \frac{\pi_{\text{test}}(y)}{\pi_{\text{train}}(y)}.$$

This approach is considered black-box as it does not require access to classifier internals, only its predicted outputs and a validation set to estimate the confusion matrix.

# F  HYPERPARAMETER TUNING FOR STRONGER BASELINES

To ensure strong and fair baselines, we perform hyperparameter tuning for each conventional ML method using the search space provided in a public implementation[9]. The search spaces are manually designed to cover commonly used ranges for each model class, including both optimization-related parameters (e.g., learning rate, max iterations) and regularization or structural options (e.g., penalty, tree depth, number of neighbors). We conduct random search over these spaces and tune the models on validation datasets that are kept separate from the final test splits. The details of the hyperparameter search spaces are provided in Table F.1.

| Model | Hyperparameter | Type | Log-scale | Range |
|---|---|---|---|---|
| Logistic Regression | max_iter | int | no | {50, 100, 200, 500, 1000} |
| | solver | categorical | no | {newton-cg, lbfgs, liblinear, sag, saga} |
| | fit_intercept | boolean | no | {True, False} |
| | penalty | categorical | no | {l1, l2, elasticnet, none} |
| | C | float | no | {0.1, 1.0, 10.0, 100.0} |
| Random Forest | n_estimators | int | no | {10, 50, 100, 200, 500} |
| | criterion | categorical | no | {gini, entropy} |
| | probability | boolean | no | {True} |
| | max_depth | int / None | no | {None, 10, 50, 100, 200} |
| | min_samples_split | int | no | {2, 5, 10} |
| | min_samples_leaf | int | no | {1, 2, 4} |
| | max_features | categorical | no | {auto, sqrt, log2} |
| SVM | C | float | no | {0.1, 1.0, 10.0, 100.0} |
| | kernel | categorical | no | {linear, poly, rbf, sigmoid} |
| | probability | boolean | no | {True} |
| | degree | int | no | {2, 3, 4, 5} |
| | gamma | categorical | no | {scale, auto} |
| MLP | max_iter | int | no | {50, 100, 200, 500, 1000} |
| | activation | categorical | no | {identity, logistic, tanh, relu} |
| | solver | categorical | no | {lbfgs, sgd, adam} |
| | alpha | float | no | {0.0001, 0.001, 0.01, 0.1} |
| | learning_rate | categorical | no | {constant, invscaling, adaptive} |
| | learning_rate_init | float | no | {0.001, 0.01, 0.1} |
| kNN | n_neighbors | int | no | {3, 5, 11, 19} |
| | weights | categorical | no | {uniform, distance} |
| | algorithm | categorical | no | {auto, ball_tree, kd_tree, brute} |
| | leaf_size | int | no | {30, 50, 100} |
| | p | int | no | {1, 2} |
| XGBoost | n_estimators | int | no | {50, 100, 200} |
| | max_depth | int | no | {6, 10, 15, 20} |
| | learning_rate | float | no | {0.001, 0.01, 0.1} |
| | subsample | float | no | {0.5, 0.6, 0.7, 0.8, 0.9, 1.0} |
| | colsample_bytree | float | no | {0.4, 0.5, ..., 1.0} |
| | colsample_bylevel | float | no | {0.4, 0.5, ..., 1.0} |
| LightGBM | n_estimators | int | no | {50, 100, 200} |
| | max_depth | int | no | {6, 10, 15, 20} |
| | learning_rate | float | no | {0.001, 0.01, 0.1} |
| | num_leaves | int | no | {31, 60, 120, 240, 480, 960} |
| | min_child_samples | int | no | {10, 20, 30, 40, 50} |
| CatBoost | iterations | int | no | {50, 100, 200} |
| | depth | int | no | {6, 8, 10} |
| | learning_rate | float | no | {0.001, 0.01, 0.1} |
| | l2_leaf_reg | float | no | {1, 3, 5, 7, 9} |

Table F.1: **Hyperparameter search spaces for each conventional ML baseline.** All hyperparameter values are tuned via random search over manually defined discrete sets.

---

[9]https://github.com/carteakey/tabpfn-eval

# G DATASET STATISTICS

We evaluate on 253 tabular datasets from OpenML (Bischl et al., 2017). Summary statistics for all datasets are provided in Table G.1, G.2, and G.3. Each dataset is described using the following attributes: the dataset name (**Name**), the total number of input features (**#Features**), the number of categorical features among them (**#Cat. Feat.**), the number of data instances (**#Instances**), the number of class labels (**#Classes**), the number of missing values (**#NaNs**), and the number of samples belonging to the smallest class (**Minority Class Size**).

| Name | #Features | #Cat. Feat. | #Instances | #Classes | #NaNs | Minority Class Size |
|---|---|---|---|---|---|---|
| pollen | 6 | 1 | 3848 | 2 | 0 | 1924 |
| Sick_numeric | 30 | 1 | 3772 | 2 | 0 | 231 |
| jungle_chess_2pcs_endgame_rat_rat | 47 | 27 | 3660 | 2 | 0 | 1605 |
| UCI_churn | 21 | 1 | 3333 | 2 | 0 | 483 |
| led24 | 25 | 25 | 3200 | 10 | 0 | 296 |
| led7 | 8 | 8 | 3200 | 10 | 0 | 270 |
| kr-vs-kp | 37 | 37 | 3196 | 2 | 0 | 1527 |
| splice | 61 | 61 | 3190 | 3 | 0 | 767 |
| space_ga | 7 | 1 | 3107 | 2 | 0 | 1541 |
| StackOverflow-polarity-train | 2 | 1 | 3097 | 3 | 0 | 842 |
| seismic-bumps | 19 | 5 | 2584 | 2 | 0 | 170 |
| ozone-level-8hr | 73 | 1 | 2534 | 2 | 0 | 160 |
| jungle_chess_2pcs_endgame_lion_lion | 47 | 27 | 2352 | 2 | 0 | 949 |
| jungle_chess_2pcs_endgame_elephant_elephant | 47 | 27 | 2351 | 2 | 0 | 1035 |
| segment | 20 | 1 | 2310 | 7 | 0 | 330 |
| Titanic | 4 | 1 | 2201 | 2 | 0 | 711 |
| quake | 4 | 1 | 2178 | 2 | 0 | 969 |
| kc1 | 22 | 1 | 2109 | 2 | 0 | 326 |
| balloon | 2 | 1 | 2001 | 2 | 0 | 482 |
| mfeat-fourier | 77 | 1 | 2000 | 10 | 0 | 200 |
| ozone-level-8hr_seed_0_nrows_2000_nclasses_10_ncols_100_stratify_True | 73 | 1 | 2000 | 2 | 0 | 126 |
| mfeat-karhunen | 65 | 1 | 2000 | 10 | 0 | 200 |
| jannis_seed_0_nrows_2000_nclasses_10_ncols_100_stratify_True | 55 | 1 | 2000 | 2 | 0 | 1000 |
| covertype_seed_0_nrows_2000_nclasses_10_ncols_100_stratify_True | 55 | 45 | 2000 | 2 | 0 | 1000 |
| first-order-theorem-proving_seed_0_nrows_2000_nclasses_10_ncols_100_stratify_True | 52 | 1 | 2000 | 6 | 0 | 159 |
| MiniBooNE_seed_0_nrows_2000_nclasses_10_ncols_100_stratify_True | 51 | 1 | 2000 | 2 | 0 | 1000 |
| KDDCup09_upselling_seed_0_nrows_2000_nclasses_10_ncols_100_stratify_True | 50 | 16 | 2000 | 2 | 0 | 1000 |
| ada_seed_0_nrows_2000_nclasses_10_ncols_100_stratify_True | 49 | 1 | 2000 | 2 | 0 | 496 |
| mfeat-zernike | 48 | 1 | 2000 | 10 | 0 | 200 |
| connect-4_seed_0_nrows_2000_nclasses_10_ncols_100_stratify_True | 43 | 43 | 2000 | 3 | 0 | 191 |
| kr-vs-kp_seed_0_nrows_2000_nclasses_10_ncols_100_stratify_True | 37 | 37 | 2000 | 2 | 0 | 956 |
| road-safety_seed_0_nrows_2000_nclasses_10_ncols_100_stratify_True | 33 | 4 | 2000 | 2 | 0 | 1000 |
| GesturePhaseSegmentationProcessed_seed_0_nrows_2000_nclasses_10_ncols_100_stratify_True | 33 | 1 | 2000 | 5 | 0 | 202 |
| PhishingWebsites_seed_0_nrows_2000_nclasses_10_ncols_100_stratify_True | 31 | 31 | 2000 | 2 | 0 | 886 |
| pol_seed_0_nrows_2000_nclasses_10_ncols_100_stratify_True | 27 | 1 | 2000 | 2 | 0 | 1000 |
| Higgs_seed_0_nrows_2000_nclasses_10_ncols_100_stratify_True | 25 | 1 | 2000 | 2 | 0 | 1000 |
| eye_movements_seed_0_nrows_2000_nclasses_10_ncols_100_stratify_True | 24 | 4 | 2000 | 2 | 0 | 1000 |
| numerai28.6_seed_0_nrows_2000_nclasses_10_ncols_100_stratify_True | 22 | 1 | 2000 | 2 | 0 | 990 |
| kc1_seed_0_nrows_2000_nclasses_10_ncols_100_stratify_True | 22 | 1 | 2000 | 2 | 0 | 309 |
| kdd_ipums_la_97-small_seed_0_nrows_2000_nclasses_10_ncols_100_stratify_True | 21 | 1 | 2000 | 2 | 0 | 1000 |
| churn_seed_0_nrows_2000_nclasses_10_ncols_100_stratify_True | 21 | 5 | 2000 | 2 | 0 | 283 |
| compass_seed_0_nrows_2000_nclasses_10_ncols_100_stratify_True | 18 | 10 | 2000 | 2 | 0 | 1000 |
| house_16H_seed_0_nrows_2000_nclasses_10_ncols_100_stratify_True | 17 | 1 | 2000 | 2 | 0 | 1000 |
| segment_seed_0_nrows_2000_nclasses_10_ncols_100_stratify_True | 17 | 1 | 2000 | 7 | 0 | 285 |
| adult_seed_0_nrows_2000_nclasses_10_ncols_100_stratify_True | 15 | 9 | 2000 | 2 | 242 | 479 |
| adult_seed_1_nrows_2000_nclasses_10_ncols_100_stratify_True | 15 | 9 | 2000 | 2 | 248 | 479 |
| adult_seed_2_nrows_2000_nclasses_10_ncols_100_stratify_True | 15 | 9 | 2000 | 2 | 279 | 479 |
| adult_seed_3_nrows_2000_nclasses_10_ncols_100_stratify_True | 15 | 9 | 2000 | 2 | 254 | 479 |
| adult_seed_4_nrows_2000_nclasses_10_ncols_100_stratify_True | 15 | 9 | 2000 | 2 | 253 | 479 |
| rl_seed_0_nrows_2000_nclasses_10_ncols_100_stratify_True | 13 | 8 | 2000 | 2 | 0 | 1000 |
| wine_seed_0_nrows_2000_nclasses_10_ncols_100_stratify_True | 12 | 1 | 2000 | 2 | 0 | 1000 |
| Click_prediction_small_seed_0_nrows_2000_nclasses_10_ncols_100_stratify_True | 12 | 7 | 2000 | 2 | 0 | 337 |
| Amazon_employee_access_seed_0_nrows_2000_nclasses_10_ncols_100_stratify_True | 10 | 10 | 2000 | 2 | 0 | 116 |
| california_seed_0_nrows_2000_nclasses_10_ncols_100_stratify_True | 9 | 1 | 2000 | 2 | 0 | 1000 |
| sf-police-incidents_seed_0_nrows_2000_nclasses_10_ncols_100_stratify_True | 9 | 6 | 2000 | 2 | 0 | 243 |
| electricity_seed_0_nrows_2000_nclasses_10_ncols_100_stratify_True | 8 | 1 | 2000 | 2 | 0 | 1000 |
| airlines_seed_0_nrows_2000_nclasses_10_ncols_100_stratify_True | 8 | 5 | 2000 | 2 | 0 | 891 |
| mfeat-morphological | 7 | 1 | 2000 | 10 | 0 | 200 |
| jungle_chess_2pcs_raw_endgame_complete_seed_0_nrows_2000_nclasses_10_ncols_100_stratify_True | 7 | 1 | 2000 | 3 | 0 | 194 |
| phoneme_seed_0_nrows_2000_nclasses_10_ncols_100_stratify_True | 6 | 1 | 2000 | 2 | 0 | 1000 |
| wilt_seed_0_nrows_2000_nclasses_10_ncols_100_stratify_True | 6 | 1 | 2000 | 2 | 0 | 108 |
| steel-plates-fault | 34 | 1 | 1941 | 2 | 0 | 673 |
| steel-plates-fault_seed_0_nrows_2000_nclasses_10_ncols_100_stratify_True | 28 | 1 | 1941 | 7 | 0 | 55 |
| GAMETES_Epistasis_2-Way_20atts_0.1H_EDM-1_1 | 21 | 21 | 1600 | 2 | 0 | 800 |
| pc3 | 38 | 1 | 1563 | 2 | 0 | 160 |
| cmc | 10 | 8 | 1473 | 3 | 0 | 333 |
| cmc_seed_0_nrows_2000_nclasses_10_ncols_100_stratify_True | 10 | 8 | 1473 | 3 | 0 | 333 |
| ibm-employee-performance | 34 | 1 | 1470 | 2 | 0 | 226 |
| pc4 | 38 | 1 | 1458 | 2 | 0 | 178 |
| pc4_seed_0_nrows_2000_nclasses_10_ncols_100_stratify_True | 38 | 1 | 1458 | 2 | 0 | 178 |
| banknote-authentication | 5 | 1 | 1372 | 2 | 0 | 610 |
| analcatdata_halloffame | 17 | 2 | 1340 | 2 | 20 | 125 |
| mofn-3-7-10 | 11 | 11 | 1324 | 2 | 0 | 292 |
| socmob | 6 | 5 | 1156 | 2 | 0 | 256 |
| parity5_plus_5 | 11 | 11 | 1124 | 2 | 0 | 557 |
| PieChart3 | 38 | 1 | 1077 | 2 | 0 | 134 |
| qsar-biodeg | 42 | 1 | 1055 | 2 | 0 | 356 |
| qsar-biodeg_seed_0_nrows_2000_nclasses_10_ncols_100_stratify_True | 42 | 1 | 1055 | 2 | 0 | 356 |
| PizzaCutter3 | 38 | 1 | 1043 | 2 | 0 | 127 |
| rmftsa_sleepdata | 3 | 1 | 1024 | 4 | 0 | 94 |
| credit-g | 21 | 14 | 1000 | 2 | 0 | 300 |
| dummy | 7 | 1 | 1000 | 2 | 0 | 273 |
| xd6 | 10 | 10 | 973 | 2 | 0 | 322 |
| tokyo1 | 45 | 3 | 959 | 2 | 0 | 346 |
| tic-tac-toe | 10 | 10 | 958 | 2 | 0 | 332 |
| Tour-and-Travels-Customer-Churn-Prediction | 7 | 5 | 954 | 2 | 60 | 224 |
| stock | 10 | 1 | 950 | 2 | 0 | 462 |
| vehicle | 19 | 1 | 846 | 4 | 0 | 199 |
| vehicle_reproduced | 19 | 1 | 846 | 4 | 0 | 199 |
| analcatdata_authorship | 71 | 1 | 841 | 4 | 0 | 55 |

Table G.1: Dataset statistics - Part 1

| Name | #Features | #Cat. Feat. | #Instances | #Classes | #NaNs | Minority Class Size |
|------|-----------|-------------|------------|----------|-------|---------------------|
| analcatdata_dmft | 5 | 5 | 797 | 6 | 0 | 123 |
| diabetes | 9 | 1 | 768 | 2 | 0 | 268 |
| blood-transfusion-service-center | 5 | 1 | 748 | 2 | 0 | 178 |
| blood-transfusion-service-center_seed_0_nrows_2000_nclasses_10_ncols_100_stratify_True | 5 | 1 | 748 | 2 | 0 | 178 |
| doa_bwin_balanced | 14 | 3 | 708 | 2 | 0 | 354 |
| PieChart1 | 38 | 1 | 705 | 2 | 0 | 61 |
| breast-w | 10 | 1 | 699 | 2 | 16 | 241 |
| credit-approval | 16 | 10 | 690 | 2 | 67 | 307 |
| credit-approval_reproduced | 16 | 10 | 690 | 2 | 67 | 307 |
| Australian | 15 | 9 | 690 | 2 | 0 | 307 |
| Australian_seed_0_nrows_2000_nclasses_10_ncols_100_stratify_True | 15 | 9 | 690 | 2 | 0 | 307 |
| disclosure_x_bias | 4 | 1 | 662 | 2 | 0 | 317 |
| disclosure_x_tampered | 4 | 1 | 662 | 2 | 0 | 327 |
| disclosure_x_noise | 4 | 1 | 662 | 2 | 0 | 329 |
| disclosure_z | 4 | 1 | 662 | 2 | 0 | 314 |
| PizzaCutter1 | 38 | 1 | 661 | 2 | 0 | 52 |
| balance-scale | 5 | 1 | 625 | 3 | 0 | 49 |
| monks-problems-2 | 7 | 7 | 601 | 2 | 0 | 206 |
| synthetic_control | 61 | 1 | 600 | 6 | 0 | 100 |
| sensory | 12 | 12 | 576 | 2 | 0 | 239 |
| wdbc | 31 | 1 | 569 | 2 | 0 | 212 |
| arsenic-female-bladder | 5 | 2 | 559 | 2 | 0 | 80 |
| monks-problems-1 | 7 | 7 | 556 | 2 | 0 | 278 |
| monks-problems-3 | 7 | 7 | 554 | 2 | 0 | 266 |
| climate-model-simulation-crashes | 21 | 1 | 540 | 2 | 0 | 46 |
| doa_bwin | 14 | 3 | 530 | 2 | 0 | 176 |
| CPMP-2015-runtime-classification | 23 | 1 | 527 | 4 | 0 | 78 |
| kc2 | 22 | 1 | 522 | 2 | 0 | 107 |
| threeOf9 | 10 | 10 | 512 | 2 | 0 | 238 |
| rmftsa_ladata | 11 | 1 | 508 | 2 | 0 | 222 |
| boston_corrected | 21 | 4 | 506 | 2 | 0 | 223 |
| boston | 14 | 2 | 506 | 2 | 0 | 209 |
| collins | 23 | 3 | 500 | 2 | 0 | 80 |
| pm10 | 8 | 1 | 500 | 2 | 0 | 246 |
| no2 | 8 | 1 | 500 | 2 | 0 | 249 |
| LED-display-domain-7digit | 8 | 1 | 500 | 10 | 0 | 37 |
| irish | 6 | 4 | 500 | 2 | 32 | 222 |
| PopularKids | 11 | 5 | 478 | 3 | 0 | 90 |
| analcatdata_apnea2 | 4 | 3 | 475 | 2 | 0 | 64 |
| analcatdata_apnea1 | 4 | 3 | 475 | 2 | 0 | 61 |
| thoracic-surgery | 17 | 14 | 470 | 2 | 0 | 70 |
| analcatdata_vineyard | 4 | 2 | 468 | 2 | 0 | 208 |
| chscase_vine2 | 3 | 1 | 468 | 2 | 0 | 212 |
| sa-heart | 10 | 2 | 462 | 2 | 0 | 160 |
| analcatdata_apnea3 | 4 | 3 | 450 | 2 | 0 | 55 |
| wholesale-customers | 9 | 2 | 440 | 2 | 0 | 142 |
| mw1 | 38 | 1 | 403 | 2 | 0 | 31 |
| user-knowledge | 6 | 1 | 403 | 5 | 0 | 24 |
| chscase_census5 | 8 | 1 | 400 | 2 | 0 | 193 |
| chscase_census4 | 8 | 1 | 400 | 2 | 0 | 194 |
| chscase_census3 | 8 | 1 | 400 | 2 | 0 | 192 |
| chscase_census2 | 8 | 1 | 400 | 2 | 0 | 197 |
| chscase_census6 | 7 | 1 | 400 | 2 | 0 | 165 |
| analcatdata_germangss | 6 | 5 | 400 | 4 | 0 | 100 |
| calendarDOW | 33 | 21 | 399 | 5 | 0 | 44 |
| autoMpg | 8 | 4 | 398 | 2 | 6 | 189 |
| vinnie | 3 | 1 | 380 | 2 | 0 | 185 |
| jEdit_4.2_4.3 | 9 | 1 | 369 | 2 | 0 | 165 |
| dermatology | 35 | 34 | 366 | 6 | 8 | 20 |
| analcatdata_draft | 5 | 3 | 366 | 2 | 1 | 32 |
| analcatdata_birthday | 4 | 3 | 365 | 2 | 30 | 53 |
| ionosphere | 35 | 1 | 351 | 2 | 0 | 126 |
| SPECTF | 45 | 1 | 349 | 2 | 0 | 95 |
| penguins | 7 | 3 | 344 | 3 | 18 | 68 |
| CastMetal1 | 38 | 1 | 327 | 2 | 0 | 42 |
| visualizing_galaxy | 5 | 1 | 323 | 2 | 0 | 148 |
| plasma_retinol | 14 | 4 | 315 | 2 | 0 | 133 |
| solar-flare | 13 | 13 | 315 | 5 | 0 | 21 |
| diggle_table_a2 | 9 | 1 | 310 | 9 | 0 | 18 |
| vertebra-column | 7 | 1 | 310 | 3 | 0 | 60 |
| haberman | 4 | 2 | 306 | 2 | 0 | 81 |
| heart-c | 14 | 8 | 303 | 2 | 7 | 138 |
| cleveland | 14 | 8 | 303 | 2 | 6 | 139 |
| cholesterol | 14 | 8 | 303 | 2 | 6 | 137 |
| cleve | 14 | 9 | 303 | 2 | 0 | 138 |
| cleveland-nominal | 8 | 8 | 303 | 5 | 0 | 13 |
| CostaMadre1 | 38 | 1 | 296 | 2 | 0 | 38 |
| Heart_disease_prediction_20 | 14 | 1 | 296 | 2 | 0 | 137 |
| breast-cancer | 10 | 10 | 286 | 2 | 9 | 85 |
| breastTumor | 10 | 9 | 286 | 2 | 9 | 120 |
| analcatdata_broadwaymult | 8 | 5 | 285 | 7 | 27 | 21 |
| mu284 | 11 | 1 | 284 | 2 | 0 | 142 |
| DiabeticMellitus | 98 | 1 | 281 | 2 | 2 | 99 |
| breast-cancer-dropped-missing-attributes-values | 10 | 10 | 277 | 2 | 0 | 81 |
| jEdit_4.0_4.2 | 9 | 1 | 274 | 2 | 0 | 134 |
| heart-statlog | 14 | 1 | 270 | 2 | 0 | 120 |
| SPECT | 23 | 23 | 267 | 2 | 0 | 55 |
| Touch2 | 11 | 1 | 265 | 8 | 0 | 27 |
| analcatdata_lawsuit | 5 | 2 | 264 | 2 | 0 | 19 |
| rmftsa_ctoarrivals | 3 | 2 | 264 | 2 | 0 | 101 |

Table G.2: Dataset statistics - Part 2

| Name | #Features | #Cat. Feat. | #Instances | #Classes | #NaNs | Minority Class Size |
|---|---|---|---|---|---|---|
| MegaWatt1 | 38 | 1 | 253 | 2 | 0 | 27 |
| bodyfat | 15 | 1 | 252 | 2 | 0 | 124 |
| qualitative-bankruptcy | 7 | 7 | 250 | 2 | 0 | 107 |
| prnn_synth | 3 | 1 | 250 | 2 | 0 | 125 |
| conference_attendance | 7 | 7 | 246 | 2 | 0 | 31 |
| chatfield_4 | 13 | 1 | 235 | 2 | 0 | 93 |
| chscase_whale | 9 | 1 | 228 | 2 | 20 | 111 |
| lungcancer_GSE31210 | 24 | 3 | 226 | 2 | 0 | 35 |
| chscase_geyser1 | 3 | 1 | 222 | 2 | 0 | 88 |
| thyroid-new | 6 | 1 | 215 | 3 | 0 | 30 |
| glass | 10 | 1 | 214 | 6 | 0 | 9 |
| prnn_fglass | 10 | 1 | 214 | 2 | 0 | 76 |
| seeds | 8 | 1 | 210 | 3 | 0 | 70 |
| biomed | 9 | 2 | 209 | 2 | 15 | 75 |
| cpu | 8 | 2 | 209 | 2 | 0 | 53 |
| machine_cpu | 7 | 1 | 209 | 2 | 0 | 56 |
| sonar | 61 | 1 | 208 | 2 | 0 | 97 |
| regime_alimentaire | 20 | 17 | 202 | 2 | 17 | 41 |
| heart-long-beach | 14 | 1 | 200 | 5 | 0 | 10 |
| pwLinear | 11 | 1 | 200 | 2 | 0 | 97 |
| prnn_crabs | 8 | 2 | 200 | 2 | 0 | 100 |
| parkinsons | 23 | 1 | 195 | 2 | 0 | 48 |
| pharynx | 11 | 10 | 195 | 2 | 2 | 74 |
| KnuggetChase3 | 40 | 1 | 194 | 2 | 0 | 36 |
| wisconsin | 33 | 1 | 194 | 2 | 0 | 90 |
| lowbwt | 10 | 8 | 189 | 2 | 0 | 90 |
| triazines | 61 | 1 | 186 | 2 | 0 | 77 |
| chscase_funds | 3 | 1 | 185 | 2 | 0 | 87 |
| planning-relax | 13 | 1 | 182 | 2 | 0 | 52 |
| Smartphone-Based_Recognition_of_Human_Activities | 68 | 2 | 180 | 6 | 0 | 30 |
| backache | 32 | 27 | 180 | 2 | 0 | 25 |
| wine | 14 | 1 | 178 | 3 | 0 | 48 |
| servo | 5 | 5 | 167 | 2 | 0 | 38 |
| robot-failures-lp5 | 91 | 1 | 164 | 5 | 0 | 21 |
| analcatdata_wildcat | 6 | 3 | 163 | 2 | 0 | 47 |
| mc2 | 40 | 1 | 161 | 2 | 0 | 52 |
| corral | 7 | 7 | 160 | 2 | 0 | 70 |
| hayes-roth | 5 | 1 | 160 | 3 | 0 | 31 |
| auto_price | 16 | 2 | 159 | 2 | 0 | 54 |
| autoPrice | 16 | 1 | 159 | 2 | 0 | 54 |
| analcatdata_gsssexsurvey | 10 | 6 | 159 | 2 | 6 | 35 |
| TuningSVMs | 81 | 1 | 156 | 2 | 0 | 54 |
| grub-damage | 9 | 7 | 155 | 4 | 0 | 19 |
| teachingAssistant | 7 | 5 | 151 | 3 | 0 | 49 |
| tae | 6 | 3 | 151 | 3 | 0 | 49 |
| iris | 5 | 1 | 150 | 3 | 0 | 50 |
| iris-example | 5 | 1 | 150 | 3 | 0 | 50 |
| sleuth_case2002 | 7 | 5 | 147 | 2 | 0 | 69 |
| kc1-top5 | 95 | 1 | 145 | 2 | 0 | 8 |
| kc1-binary | 95 | 1 | 145 | 2 | 0 | 60 |
| newton_hema | 4 | 2 | 140 | 2 | 0 | 70 |
| veteran | 8 | 5 | 137 | 2 | 0 | 43 |
| analcatdata_boxing2 | 4 | 4 | 132 | 2 | 0 | 61 |
| analcatdata_seropositive | 4 | 2 | 132 | 2 | 0 | 46 |
| transplant | 4 | 1 | 131 | 2 | 0 | 48 |
| datatrieve | 9 | 1 | 130 | 2 | 0 | 11 |
| visualizing_livestock | 3 | 2 | 130 | 5 | 0 | 26 |
| humandevel | 2 | 1 | 130 | 2 | 0 | 65 |
| mux6 | 7 | 7 | 128 | 2 | 0 | 64 |
| MindCave2 | 40 | 1 | 125 | 2 | 0 | 44 |
| fruitfly | 5 | 3 | 125 | 2 | 0 | 49 |
| KungChi3 | 40 | 1 | 123 | 2 | 0 | 16 |
| heart-switzerland | 13 | 1 | 123 | 5 | 0 | 5 |
| ar1 | 30 | 1 | 121 | 2 | 0 | 9 |
| analcatdata_boxing1 | 4 | 4 | 120 | 2 | 0 | 42 |
| rabe_266 | 3 | 1 | 120 | 2 | 0 | 57 |
| robot-failures-lp4 | 91 | 1 | 117 | 3 | 0 | 21 |
| visualizing_environmental | 4 | 1 | 111 | 2 | 0 | 53 |
| cloud | 8 | 2 | 108 | 2 | 0 | 32 |
| analcatdata_michiganacc | 4 | 3 | 108 | 2 | 0 | 48 |
| ar4 | 30 | 1 | 107 | 2 | 0 | 20 |
| molecular-biology_promoters | 58 | 58 | 106 | 2 | 0 | 53 |
| breast-tissue | 10 | 1 | 106 | 6 | 0 | 14 |
| ar6 | 30 | 1 | 101 | 2 | 0 | 15 |
| zoo | 17 | 16 | 101 | 7 | 0 | 4 |
| fertility | 10 | 1 | 100 | 2 | 0 | 12 |
| analcatdata_creditscore | 7 | 4 | 100 | 2 | 0 | 27 |
| blogger | 6 | 6 | 100 | 2 | 0 | 32 |
| analcatdata_chlamydia | 4 | 4 | 100 | 2 | 0 | 19 |
| analcatdata_neavote | 3 | 2 | 100 | 2 | 0 | 7 |

Table G.3: Dataset statistics - Part 3

# H    K-MEANS CLUSTERING FOR DATASET SELECTION

Table H.1 reports the extended results of our K-means clustering-based training set selection under different numbers of clusters $K \in \{3, 5, 10\}$, where a proportion ($P \in \{0.05, 0.10, 0.20\}$) of samples is drawn from each cluster. Across all settings, our method demonstrates stable performance regardless of $K$, confirming its robustness when applied with clustering-based selection.

| $K$ | $P$ | Methods | w/o shift | Shift strength ($\beta$) | | | | | | Avg. |
|---|---|---|---|---|---|---|---|---|---|---|
| | | | | 0.0 | 0.1 | 0.5 | 1.0 | 2.0 | 5.0 | |
| 3 | 0.05 | TabPFN-v2 | 0.668 | 0.596 | 0.591 | 0.548 | 0.500 | 0.439 | 0.408 | 0.513 |
| | | DistPFN | 0.661 | 0.622 | 0.614 | 0.579 | 0.532 | 0.454 | 0.428 | 0.538 |
| | | DistPFN-T | 0.657 | **0.625** | **0.616** | **0.588** | **0.540** | **0.459** | **0.433** | **0.543** |
| | 0.10 | TabPFN-v2 | 0.699 | 0.626 | 0.632 | 0.570 | 0.528 | 0.465 | 0.424 | 0.541 |
| | | DistPFN | 0.692 | 0.641 | 0.653 | 0.620 | 0.564 | 0.498 | 0.450 | 0.573 |
| | | DistPFN-T | 0.687 | **0.643** | **0.657** | **0.628** | **0.569** | **0.504** | **0.454** | **0.584** |
| | 0.20 | TabPFN-v2 | 0.732 | 0.673 | 0.668 | 0.626 | 0.576 | 0.505 | 0.468 | 0.591 |
| | | DistPFN | 0.727 | 0.692 | 0.688 | 0.669 | 0.614 | 0.556 | 0.509 | 0.639 |
| | | DistPFN-T | 0.722 | **0.691** | **0.692** | **0.674** | **0.620** | **0.568** | **0.516** | **0.661** |
| 5 | 0.05 | TabPFN-v2 | 0.676 | 0.605 | 0.606 | 0.550 | 0.503 | 0.459 | 0.429 | 0.529 |
| | | DistPFN | 0.673 | 0.628 | 0.630 | 0.587 | 0.534 | 0.487 | 0.453 | 0.561 |
| | | DistPFN-T | 0.672 | **0.629** | **0.634** | **0.594** | **0.539** | **0.493** | **0.460** | **0.565** |
| | 0.10 | TabPFN-v2 | 0.699 | 0.631 | 0.644 | 0.586 | 0.540 | 0.485 | 0.446 | 0.569 |
| | | DistPFN | 0.696 | 0.654 | 0.665 | 0.624 | 0.583 | 0.528 | 0.475 | 0.609 |
| | | DistPFN-T | 0.693 | **0.655** | **0.670** | **0.630** | **0.591** | **0.538** | **0.483** | **0.620** |
| | 0.20 | TabPFN-v2 | 0.732 | 0.670 | 0.679 | 0.628 | 0.582 | 0.523 | 0.481 | 0.618 |
| | | DistPFN | 0.736 | 0.687 | 0.697 | 0.662 | 0.625 | 0.576 | 0.521 | 0.645 |
| | | DistPFN-T | 0.731 | **0.690** | **0.698** | **0.670** | **0.631** | **0.584** | **0.531** | **0.667** |
| 10 | 0.05 | TabPFN-v2 | 0.708 | 0.644 | 0.639 | 0.591 | 0.547 | 0.505 | 0.460 | 0.554 |
| | | DistPFN | 0.706 | 0.659 | 0.662 | 0.627 | 0.586 | 0.548 | 0.493 | 0.589 |
| | | DistPFN-T | 0.701 | **0.662** | **0.667** | **0.640** | **0.601** | **0.562** | **0.504** | **0.605** |
| | 0.10 | TabPFN-v2 | 0.727 | 0.663 | 0.664 | 0.617 | 0.582 | 0.534 | 0.481 | 0.620 |
| | | DistPFN | 0.723 | 0.679 | 0.685 | 0.650 | 0.618 | 0.577 | 0.515 | 0.642 |
| | | DistPFN-T | 0.718 | **0.685** | **0.691** | **0.656** | **0.629** | **0.588** | **0.527** | **0.652** |
| | 0.20 | TabPFN-v2 | 0.749 | 0.697 | 0.689 | 0.651 | 0.619 | 0.561 | 0.510 | 0.638 |
| | | DistPFN | 0.749 | 0.713 | 0.711 | 0.682 | 0.663 | 0.610 | 0.553 | 0.676 |
| | | DistPFN-T | 0.748 | **0.716** | **0.715** | **0.689** | **0.670** | **0.620** | **0.563** | **0.688** |

Table H.1: **K-means-based training dataset selection.** Our method remains effective when training subsets are selected by clustering the data and sampling a percentage ($P$) of samples from each of $K$ clusters.

# I    APPLICATION TO LOCALPFN

Table I.1 provides the full results for LoCalPFN under different values of $k$ across six $\beta$ values. The results confirm that our methods yield consistent improvements regardless of the choice of $k$, demonstrating robustness of the approach.

| $k$ | Methods | Shift strength ($\beta$) | | | | | | Avg. |
|---|---|---|---|---|---|---|---|---|
| | | 0.0 | 0.1 | 0.5 | 1.0 | 2.0 | 5.0 | |
| 3 | LoCalPFN | 0.789 | 0.787 | 0.774 | 0.758 | 0.711 | 0.679 | 0.750 |
| | + DistPFN | 0.794 | 0.794 | 0.792 | 0.786 | 0.772 | 0.752 | 0.782 |
| | + DistPFN-T | **0.794** | **0.794** | **0.794** | **0.790** | **0.779** | **0.759** | **0.785** |
| 5 | LoCalPFN | 0.792 | 0.791 | 0.785 | 0.775 | 0.744 | 0.714 | 0.767 |
| | + DistPFN | 0.794 | 0.795 | 0.793 | 0.790 | 0.777 | 0.766 | 0.786 |
| | + DistPFN-T | **0.795** | **0.796** | **0.795** | **0.794** | **0.784** | **0.770** | **0.789** |
| 10 | LoCalPFN | 0.794 | 0.792 | 0.786 | 0.778 | 0.752 | 0.720 | 0.770 |
| | + DistPFN | 0.796 | 0.795 | 0.793 | 0.791 | 0.779 | 0.768 | 0.787 |
| | + DistPFN-T | **0.797** | **0.797** | **0.796** | **0.794** | **0.785** | **0.774** | **0.789** |
| 20 | LoCalPFN | 0.794 | 0.793 | 0.788 | 0.778 | 0.753 | 0.719 | 0.771 |
| | + DistPFN | 0.797 | 0.796 | 0.794 | 0.790 | 0.782 | 0.770 | 0.788 |
| | + DistPFN-T | **0.798** | **0.797** | **0.796** | **0.794** | **0.787** | **0.776** | **0.791** |

Table I.1: **Application to LoCalPFN.** DistPFN and DistPFN-T applied to LoCalPFN show consistent improvements across varying numbers of neighbors ($k$).

## J    COMPARISON WITH METHODS FOR LABEL SHIFT CORRECTION

To demonstrate the effectiveness of our approach, we compare it with classical methods for handling label shift by rescaling classifier outputs, which typically require estimating the test distribution: EM-based Estimation (EME) (Saerens et al., 2002) and Black-box Estimation (BBE) (Lipton et al., 2018). Table J.1 presents the results, showing that our method is effective without requiring estimation of the test prior.

| Methods | w/o shift | Shift strength ($\beta$) | | | | | | |
|---|---|---|---|---|---|---|---|---|
| | | 0.0 | 0.1 | 0.5 | 1.0 | 2.0 | 5.0 | Avg. |
| LoCalPFN | **0.816** | 0.794 | 0.793 | 0.788 | 0.778 | 0.753 | 0.719 | 0.771 |
| + EME | 0.801 | 0.792 | 0.790 | 0.786 | 0.785 | 0.778 | 0.769 | 0.783 |
| + BBE | 0.805 | **0.798** | 0.795 | 0.792 | 0.789 | 0.782 | 0.770 | 0.787 |
| + DistPFN | **0.816** | 0.797 | 0.796 | 0.794 | 0.790 | 0.782 | 0.770 | 0.788 |
| + DistPFN-T | **0.816** | **0.798** | **0.797** | **0.796** | **0.794** | **0.787** | **0.776** | **0.791** |
| TabICL | **0.806** | 0.783 | 0.781 | 0.770 | 0.747 | 0.704 | 0.664 | 0.742 |
| + EME | 0.798 | 0.776 | 0.776 | 0.770 | 0.769 | 0.761 | 0.747 | 0.766 |
| + BBE | 0.802 | 0.783 | 0.785 | 0.780 | 0.774 | 0.754 | 0.734 | 0.768 |
| + DistPFN | **0.806** | 0.786 | 0.786 | 0.781 | 0.776 | 0.763 | 0.746 | 0.773 |
| + DistPFN-T | **0.806** | **0.786** | **0.786** | **0.783** | **0.780** | **0.771** | **0.755** | **0.777** |
| TabPFN-v2 | **0.818** | 0.797 | 0.796 | 0.790 | 0.782 | 0.759 | 0.727 | 0.775 |
| + EME | 0.801 | 0.793 | 0.793 | 0.790 | 0.787 | 0.783 | 0.768 | 0.786 |
| + BBE | 0.805 | **0.799** | 0.797 | **0.797** | 0.791 | 0.783 | 0.768 | 0.789 |
| + DistPFN | **0.818** | **0.799** | 0.797 | 0.795 | 0.791 | 0.783 | 0.769 | 0.789 |
| + DistPFN-T | **0.818** | **0.799** | **0.798** | **0.797** | **0.796** | **0.789** | **0.775** | **0.792** |

Figure J.1: Comparison with other label shift methods.

## K PREDICTED DISTRIBUTION OF SINGLE VS. MULTIPLE INSTANCES

As TabPFN produces identical predictions whether test instances are evaluated individually or in batches, DistPFN and DistPFN-T can adjust based on either 1) the prediction of a *single* instance or 2) the average prediction across *multiple* instances. As shown in Table K.1, both choices consistently improve TabPFN-v2 (Hollmann et al., 2025), averaged across six $\beta$s for *w/ shift*, demonstrating robustness to the choice of distribution source.

| | Pred. distn. | w/o shift | Shift strength ($\beta$) | | | | | | |
|---|---|---|---|---|---|---|---|---|---|
| | | | 0.0 | 0.1 | 0.5 | 1.0 | 2.0 | 5.0 | Avg. |
| TabPFN-v2 | - | 0.818 | 0.797 | 0.796 | 0.790 | 0.782 | 0.759 | 0.727 | 0.775 |
| + DistPFN | Single | **0.818** | 0.797 | 0.796 | **0.795** | **0.793** | **0.784** | **0.770** | **0.789** |
| | Multiple | **0.818** | **0.799** | **0.797** | **0.795** | 0.791 | 0.783 | **0.770** | **0.789** |
| + DistPFN-T | Single | **0.818** | 0.797 | 0.797 | 0.796 | 0.795 | 0.788 | 0.773 | 0.791 |
| | Multiple | **0.818** | **0.799** | **0.798** | **0.797** | **0.796** | **0.789** | **0.775** | **0.792** |

Table K.1: **Predicted distributions: Single vs. Multiple.** The proposed methods consistently improves TabPFN-v2 regardless of whether the adjustment is based on single or aggregated distribution.

## L  OTHER METRICS

Table L.1 reports the comparison of our methods and baselines under $\beta = 2$ in terms of ROC-AUC, demonstrating the effectiveness of our method. The results demonstrate that our method shows nearly the same values as the backbone, as the adjustment only rescales predicted probabilities without altering their order.

| | Methods | w/o shift | Shift strength ($\beta$) | | | | | | Avg. |
|---|---|---|---|---|---|---|---|---|---|
| | | | 0.0 | 0.1 | 0.5 | 1.0 | 2.0 | 5.0 | |
| Machine Learning | LogReg. | $0.813_{\pm 0.002}$ | $0.789_{\pm 0.002}$ | $0.789_{\pm 0.002}$ | $0.790_{\pm 0.002}$ | $0.788_{\pm 0.002}$ | $0.784_{\pm 0.003}$ | $0.777_{\pm 0.002}$ | 0.786 |
| | + HPO | $0.817_{\pm 0.002}$ | $0.806_{\pm 0.001}$ | $0.806_{\pm 0.001}$ | $0.805_{\pm 0.002}$ | $0.803_{\pm 0.002}$ | $0.797_{\pm 0.001}$ | $0.791_{\pm 0.001}$ | 0.801 |
| | SVM | $0.815_{\pm 0.002}$ | $0.744_{\pm 0.004}$ | $0.747_{\pm 0.005}$ | $0.750_{\pm 0.004}$ | $0.744_{\pm 0.004}$ | $0.733_{\pm 0.005}$ | $0.725_{\pm 0.006}$ | 0.741 |
| | + HPO | $0.840_{\pm 0.002}$ | $0.804_{\pm 0.003}$ | $0.804_{\pm 0.003}$ | $0.800_{\pm 0.001}$ | $0.799_{\pm 0.004}$ | $0.794_{\pm 0.002}$ | $0.785_{\pm 0.002}$ | 0.798 |
| | MLP | $0.821_{\pm 0.003}$ | $0.747_{\pm 0.002}$ | $0.750_{\pm 0.003}$ | $0.746_{\pm 0.005}$ | $0.735_{\pm 0.004}$ | $0.719_{\pm 0.002}$ | $0.702_{\pm 0.004}$ | 0.733 |
| | + HPO | $0.849_{\pm 0.003}$ | $0.799_{\pm 0.001}$ | $0.799_{\pm 0.002}$ | $0.796_{\pm 0.001}$ | $0.788_{\pm 0.003}$ | $0.781_{\pm 0.003}$ | $0.772_{\pm 0.002}$ | 0.789 |
| | $k$NN | $0.789_{\pm 0.001}$ | $0.728_{\pm 0.003}$ | $0.728_{\pm 0.004}$ | $0.727_{\pm 0.004}$ | $0.722_{\pm 0.004}$ | $0.709_{\pm 0.003}$ | $0.693_{\pm 0.003}$ | 0.718 |
| | + HPO | $0.828_{\pm 0.002}$ | $0.775_{\pm 0.002}$ | $0.775_{\pm 0.003}$ | $0.773_{\pm 0.002}$ | $0.768_{\pm 0.002}$ | $0.756_{\pm 0.001}$ | $0.742_{\pm 0.002}$ | 0.765 |
| | Random Forest | $0.836_{\pm 0.003}$ | $0.824_{\pm 0.003}$ | $0.823_{\pm 0.003}$ | $0.821_{\pm 0.003}$ | $0.818_{\pm 0.002}$ | $0.812_{\pm 0.001}$ | $0.802_{\pm 0.001}$ | 0.817 |
| | + HPO | $0.849_{\pm 0.003}$ | $0.836_{\pm 0.003}$ | $0.835_{\pm 0.003}$ | $0.834_{\pm 0.003}$ | $0.832_{\pm 0.002}$ | $0.826_{\pm 0.001}$ | $0.818_{\pm 0.002}$ | 0.830 |
| | LightGBM | $0.824_{\pm 0.002}$ | $0.813_{\pm 0.001}$ | $0.812_{\pm 0.001}$ | $0.809_{\pm 0.002}$ | $0.805_{\pm 0.003}$ | $0.797_{\pm 0.002}$ | $0.785_{\pm 0.003}$ | 0.805 |
| | + HPO | $0.845_{\pm 0.003}$ | $0.776_{\pm 0.009}$ | $0.767_{\pm 0.006}$ | $0.781_{\pm 0.003}$ | $0.774_{\pm 0.010}$ | $0.775_{\pm 0.009}$ | $0.779_{\pm 0.004}$ | 0.775 |
| | CatBoost | $0.847_{\pm 0.003}$ | $0.833_{\pm 0.003}$ | $0.832_{\pm 0.002}$ | $0.830_{\pm 0.003}$ | $0.827_{\pm 0.002}$ | $0.820_{\pm 0.002}$ | $0.810_{\pm 0.002}$ | 0.825 |
| | + HPO | $0.843_{\pm 0.003}$ | $0.833_{\pm 0.003}$ | $0.832_{\pm 0.002}$ | $0.830_{\pm 0.003}$ | $0.827_{\pm 0.002}$ | $0.819_{\pm 0.002}$ | $0.810_{\pm 0.001}$ | 0.825 |
| Deep Learning — Non-found. | FT-Transformer | $0.821_{\pm 0.003}$ | $0.818_{\pm 0.003}$ | $0.819_{\pm 0.002}$ | $0.816_{\pm 0.003}$ | $0.812_{\pm 0.002}$ | $0.795_{\pm 0.003}$ | $0.771_{\pm 0.002}$ | 0.805 |
| | TabM | $0.824_{\pm 0.003}$ | $0.824_{\pm 0.003}$ | $0.824_{\pm 0.002}$ | $0.823_{\pm 0.003}$ | $0.821_{\pm 0.001}$ | $0.808_{\pm 0.003}$ | $0.791_{\pm 0.002}$ | 0.815 |
| | TabulaRNN | $0.774_{\pm 0.003}$ | $0.699_{\pm 0.003}$ | $0.684_{\pm 0.002}$ | $0.641_{\pm 0.003}$ | $0.585_{\pm 0.009}$ | $0.522_{\pm 0.011}$ | $0.465_{\pm 0.008}$ | 0.599 |
| | MambaTab | $0.743_{\pm 0.005}$ | $0.629_{\pm 0.006}$ | $0.603_{\pm 0.004}$ | $0.525_{\pm 0.002}$ | $0.466_{\pm 0.010}$ | $0.430_{\pm 0.005}$ | $0.394_{\pm 0.002}$ | 0.508 |
| | RealMLP | $0.821_{\pm 0.002}$ | $0.805_{\pm 0.003}$ | $0.806_{\pm 0.002}$ | $0.807_{\pm 0.002}$ | $0.804_{\pm 0.003}$ | $0.795_{\pm 0.000}$ | $0.781_{\pm 0.004}$ | 0.800 |
| Deep Learning — Foundation | LoCalPFN | $0.858_{\pm 0.002}$ | $0.842_{\pm 0.002}$ | $0.840_{\pm 0.001}$ | $0.839_{\pm 0.000}$ | $0.836_{\pm 0.000}$ | $0.830_{\pm 0.000}$ | $0.826_{\pm 0.001}$ | 0.836 |
| | + DistPFN | $0.858_{\pm 0.002}$ | $0.842_{\pm 0.002}$ | $0.840_{\pm 0.002}$ | $0.839_{\pm 0.001}$ | $0.836_{\pm 0.001}$ | $0.830_{\pm 0.001}$ | $0.826_{\pm 0.002}$ | 0.836 |
| | + DistPFN-T | $0.858_{\pm 0.002}$ | $0.842_{\pm 0.002}$ | $0.840_{\pm 0.002}$ | $0.839_{\pm 0.001}$ | $0.837_{\pm 0.001}$ | $0.830_{\pm 0.001}$ | $0.826_{\pm 0.001}$ | 0.836 |
| | TabICL | $0.845_{\pm 0.003}$ | $0.832_{\pm 0.003}$ | $0.832_{\pm 0.001}$ | $0.830_{\pm 0.002}$ | $0.826_{\pm 0.002}$ | $0.821_{\pm 0.002}$ | $0.813_{\pm 0.003}$ | 0.826 |
| | + DistPFN | $0.845_{\pm 0.003}$ | $0.832_{\pm 0.003}$ | $0.832_{\pm 0.001}$ | $0.830_{\pm 0.002}$ | $0.826_{\pm 0.002}$ | $0.821_{\pm 0.002}$ | $0.814_{\pm 0.003}$ | 0.826 |
| | + DistPFN-T | $0.845_{\pm 0.003}$ | $0.832_{\pm 0.003}$ | $0.832_{\pm 0.001}$ | $0.830_{\pm 0.002}$ | $0.826_{\pm 0.002}$ | $0.821_{\pm 0.002}$ | $0.814_{\pm 0.003}$ | 0.826 |
| | TabPFN-v2 | $0.859_{\pm 0.002}$ | $0.843_{\pm 0.002}$ | $0.842_{\pm 0.003}$ | $0.841_{\pm 0.002}$ | $0.838_{\pm 0.002}$ | $0.833_{\pm 0.001}$ | $0.826_{\pm 0.002}$ | 0.837 |
| | + DistPFN | $0.859_{\pm 0.002}$ | $0.843_{\pm 0.002}$ | $0.843_{\pm 0.003}$ | $0.841_{\pm 0.002}$ | $0.838_{\pm 0.002}$ | $0.833_{\pm 0.001}$ | $0.826_{\pm 0.003}$ | 0.837 |
| | + DistPFN-T | $0.859_{\pm 0.002}$ | $0.843_{\pm 0.002}$ | $0.842_{\pm 0.003}$ | $0.841_{\pm 0.002}$ | $0.838_{\pm 0.002}$ | $0.833_{\pm 0.001}$ | $0.826_{\pm 0.003}$ | 0.837 |

Table L.1: Tabular classification results: ROC-AUC comparisons.

