# OpenReview forum: "DistPFN: Test-Time Posterior Adjustment for Tabular Foundation Models under Label Shift"
_ICLR.cc/2026/Conference — ICLR 2026 Conference Withdrawn Submission_

### Official Review · Reviewer_cTey · 2025-10-16

**Soundness:** 2
**Presentation:** 1
**Contribution:** 2
**Rating:** 4
**Confidence:** 4

**Summary:**

This paper introduces DISTPFN, a test-time adaptation method that solves the class bias through prediction adjustment. The proposed DistPFN introduces a test-time posterior adjustment that rescales predicted probabilities by reducing reliance on the training prior and amplifying the model’s posterior signal—without altering the architecture or requiring retraining. An extended variant, DistPFN-T, further applies temperature scaling to dynamically modulate the adjustment strength based on prior–posterior discrepancy. The experiment shows effectiveness.

**Strengths:**

1. This paper studies a novel problem;
2. Empirical results show improvements compared to baselines over multiple datasets.

**Weaknesses:**

1. This paper is hard to follow, for example, in Table 1 and 2, the meaning of the numbers is unclear. The authors should at least clarify the meaning of the numbers in the tables—whether they represent prediction accuracy, estimated distributions, or the proportion of samples predicted as a specific class. The lack of explanation severely affects the readability of the paper.

2. The method lacks novelty and clear motivation. In Eq. (2), the authors use the predictions of TabPFN and the training prior to adjust the predicted probabilities. However, from both the equation and the subsequent explanation, the numerator in the second term does not appear to reliably estimate the distribution, since TabPFN supports single-instance testing, and few samples can lead to significant estimation bias. Therefore, I question the soundness of Eq. (2) and the proposed method. The authors should clarify this issue, and also consider improving the readability of mathematical notation, as many equations are difficult to interpret. Moreover, similar approaches have been widely adopted in prior test-time adaptation (TTA) works [1][2][3], which limits the novelty of this paper.

3. Many studies have introduced real datasets with label shift, such as TableShift[4]. Why did the authors not conduct experiments on such datasets, which are closer to realistic scenarios, instead of creating their own samples?

4. The related work section is severely insufficient. Although the paper claims to propose a test-time adaptation method, it lacks both a discussion of related TTA research and comparisons with existing methods, neglecting an important body of work. In addition, the analysis of TabPFN itself is incomplete—for example, the Drift-Resilient TabPFN[5] method that addresses OOD issues is not discussed.

5. Formatting issue: According to the ICLR template, the captions of tables should appear above the tables.

6. Layout issue: Figures are clustered together, and readers often have to scroll through multiple pages to locate the corresponding figures and tables.

Ref:

[1] Boudiaf, M., Mueller, R., Ben Ayed, I., & Bertinetto, L. (2022). Parameter-free online test-time adaptation. In Proceedings of the IEEE/CVF Conference on Computer Vision and Pattern Recognition (pp. 8344-8353).

[2] Kim, C., Kim, T., Woo, S., Yang, J. Y., & Yang, E. (2024). Adaptable: Test-time adaptation for tabular data via shift-aware uncertainty calibrator and label distribution handler. arXiv preprint arXiv:2407.10784.

[3] Zhou, Z., Yu, K. Y., Guo, L. Z., & Li, Y. F. (2025, April). Fully test-time adaptation for tabular data. In Proceedings of the AAAI Conference on Artificial Intelligence (Vol. 39, No. 21, pp. 23027-23035).

[4] Gardner, J., Popovic, Z., & Schmidt, L. (2023). Benchmarking distribution shift in tabular data with tableshift. Advances in Neural Information Processing Systems, 36, 53385-53432.

[5] Helli, K., Schnurr, D., Hollmann, N., Müller, S., & Hutter, F. (2024). Drift-resilient tabPFN: In-context learning temporal distribution shifts on tabular data. Advances in Neural Information Processing Systems, 37, 98742-98781.

**Questions:**

See Weaknesses above

---

### Official Review · Reviewer_Jtvb · 2025-10-27

**Soundness:** 3
**Presentation:** 3
**Contribution:** 3
**Rating:** 4
**Confidence:** 4

**Summary:**

This paper proposes DistPFN, a novel and simple test-time adaptation method that adjusts TabPFN’s output distribution based on the ratio between the predicted posterior and the prior representing the class distribution of the training dataset. This paper further introduce DistPFN-T, which extends this approach by applying temperature scaling to adaptively control the strength of adjustment according to the degree of distributional mismatch between the prior and the posterior.

**Strengths:**

1.Introduces DistPFN and DistPFN-T, simple yet effective test-time adaptation methods for label shift in tabular foundation models.

2.Demonstrates robust improvements across 253 datasets, especially under severe label shift, without modifying model parameters or requiring extra training.

3.Provides a theoretically grounded approach, connecting Bayesian inference and classical label shift correction.

**Weaknesses:**

1.The work lacks innovation. This processing method isn't first proposed in this paper. This processing method is not limited to ICL models; any model with logits output can use it. The paper lacks corresponding result.

2.The paper uses sampling to implement label shift, which may not simulate complex real-world scenarios. You can refer to datasets in TabReD for a better approach.

3.I‘m skeptical about the effectiveness of the temperature value. Suppose the training data has a distribution of 7:3, and different label shifts in the test data (becoming more balanced or more imbalanced) could correspond to the same temperature value. Can the same temperature adapt to different label shifts?

**Questions:**

Follew the weakness.

---

### Official Review · Reviewer_gH2E · 2025-10-31

**Soundness:** 1
**Presentation:** 2
**Contribution:** 1
**Rating:** 0
**Confidence:** 4

**Summary:**

The paper proposes a method to rescale the prediction probabilities of a tabular predictive model using the distribution of the classes from training and test data to improve the performance under class imbalance and so-called label shift. The authors compare their method on many datasets with many different models while proposing it as an extension of TabPFN (or other PFN-based tabular foundation models).

**Strengths:**

The strengths of the paper are its extensive collection of datasets and general experiments. Moreover, the presentation and writing are mostly clear.

**Weaknesses:**

The work has apparent fundamental flaws and an extremely misleading framing. In its current state, the framing of the results and method is highly questionable.

To clarify the fundamental flaw, observe Table 3 and Table L.1. Table 3 presents the results for accuracy (a threshold metric based on a decision boundary/threshold) while Table L.1 presents the results for ROC AUC. ROC AUC is a threshold-independent metric, such that rescaling the prediction does not affect the results. Moreover, its scores are not affected by a threshold that otherwise needs to be tuned.  All results but Table L.1 use accuracy to argue for better or worse performance. We clearly observe that the newly proposed method has no effect, as the authors also state in the caption, due to ROC AUC only scoring based on order. In other words, the presented work shows almost all its results for a metric that requires threshold tuning or calibration, but applies neither properly in its experiments anywhere else in the paper. It is very likely that most of the experiments would look very different if the baselines or any method had been correctly calibrated or threshold tuned per training data set (i.e., per shift).

Moreover, the author position their entire work as some kind of extension of a PFN-based tabular foundation model, while nothing about their method is specific to a PFN. The method can be applied to any tabular model returning prediction probabilities. The author claims in Line 193 ff that only ICL-based methods have access to the training data at inference time, such that no other model can use their method **since one could not compute the distribution of classes in the training data during the prediction**. This is clearly false; a simple counterexample is kNN. Moreover, any method could simply cache the training data. Besides, one would not need to cache the training data because one could simply compute the distribution while training... on the training data.

Finally, the authors compare only in one table to two real baselines (other calibration methods), and this does not include threshold calibration (https://scikit-learn.org/stable/modules/classification_threshold.html) or proper temperature scaling (e.g., https://arxiv.org/abs/2501.19195). So we have no indication of the performance of the method compared to real baselines. Lastly, the proposed method utilizes test samples (in batches or individually), which constitutes transductive learning, while all other compared methods perform inductive learning (i.e., they never see the test samples). Thus, transductive learning baseline are also missing.

There are a few more flaws in the paper, I will not go into detail as the flaws above make the work unacceptable anyhow. I list a short overview below:
* The work claims that TabPFN has a bias for imbalanced data or label shift, while all evidence for this supposed bias only stems from the evaluation metric.
* The concept of label shift introduced in this work is almost identical to the concept of class imbalance. Moreover, the way the shifts are created in this work does not seem to create any representative tasks for real-world tabular prediction problems.
* The related work consists of almost no relevant work and instead reads like a background section with almost no relevant information for the contributions of the paper itself.
* There is no information on how the datasets were collected, the collection contains duplicates, and the data is generally dirty, which is not useful for the benchmark at hand. Moreover, the validation setup used is highly outdated. In lines 309ff, the paper cites Hollmann et al. 2023 and 2025 as motivation for their setup, while ignoring that Hollmann et al. 2025 used a different setup to address the problems of Hollmann et al. 2023 and the setup used in this work.
* If one uses HPO, it is trivial to tune the threshold of a method.

**Questions:**

N/A

---

### Official Review · Reviewer_DFHS · 2025-11-10

**Soundness:** 3
**Presentation:** 3
**Contribution:** 3
**Rating:** 6
**Confidence:** 3

**Summary:**

This work introduces two methods to postprocess training predictions of the tabular prediction method TabPFN to better handle potential label shift, so having a different label distribution on the test than on the training set. Both methods amplify the prediction's difference to the training prior label distribution.  The authors evaluate their methods on 253 OpenML datasets, simulating varying degrees of label shift via inverse-frequency-based oversampling of training data while keeping the test set unchanged. The proposed approaches maintain accuracy in unshifted settings and significantly outperform existing label-shift correction baselines when shift is introduced.

**Strengths:**

* Clear identification of an important limitation in existing PFN models.
* Simple and well-motivated adjustment method that requires no retraining.
* Comprehensive experimental evaluation across >250 datasets and PFN variants.
* Maintains accuracy in standard settings while improving robustness under shift.

**Weaknesses:**

* The specific mathematical form of the adjustment (e.g., squaring the posterior in DistPFN, the cross-entropy-based temperature in DistPFN-T) seems somewhat ad hoc and lacks a strong theoretical justification.
* The evaluation simulates label shift synthetically via oversampling. It would strengthen the paper to include naturally shifted datasets, then one could also compare to simple oversampling of the training data to achieve equal label distributions as an alternative correction method. Maybe one could simulate this scenario by undersampling the train set instead of oversampling, that would be interesting.

**Questions:**

- In Algorithm 1, is **α** a scalar (equal to 1 when `method=="tabpfn"`) but a vector in the DistPFN and DistPFN-T cases?
- Is **τ** (tau) indeed a scalar temperature shared across all classes, computed from the cross-entropy between predicted and training priors?

---

### Note · Authors · 2025-11-15

I have read and agree with the venue's withdrawal policy on behalf of myself and my co-authors.